EMBO
Molecular Medicine

# A streptococcal lipid toxin induces membrane permeabilization and pyroptosis leading to fetal injury

Christopher Whidbey[1,2], Jay Vornhagen[1,2,†], Claire Gendrin[1,†], Erica Boldenow[1,†], Jenny Mae Samson[3,†], Kenji Doering[3], Lisa Ngo[1], Ejiofor A D Ezekwe Jr.[4], Jens H Gundlach[3], Michal A Elovitz[5], Denny Liggitt[6], Joseph A Duncan[4], Kristina M Adams Waldorf[7] & Lakshmi Rajagopal[1,2,*]

## Abstract

Group B streptococci (GBS) are Gram-positive bacteria that cause infections *in utero* and in newborns. We recently showed that the GBS pigment is hemolytic and increased pigment production promotes bacterial penetration of human placenta. However, mechanisms utilized by the hemolytic pigment to induce host cell lysis and the consequence on fetal injury are not known. Here, we show that the GBS pigment induces membrane permeability in artificial lipid bilayers and host cells. Membrane defects induced by the GBS pigment trigger $K^+$ efflux leading to osmotic lysis of red blood cells or pyroptosis in human macrophages. Macrophages lacking the NLRP3 inflammasome recovered from pigment-induced cell damage. In a murine model of *in utero* infection, hyperpigmented GBS strains induced fetal injury in both an NLRP3 inflammasome-dependent and NLRP3 inflammasome-independent manner. These results demonstrate that the dual mechanism of action of the bacterial pigment/lipid toxin leading to hemolysis or pyroptosis exacerbates fetal injury and suggest that preventing both activities of the hemolytic lipid is likely critical to reduce GBS fetal injury and preterm birth.

**Keywords** cell death; Group B streptococcus; hemolytic pigment; inflammasome; preterm birth

**Subject Categories** Microbiology, Virology & Host Pathogen Interaction; Urogenital System

## Introduction

Preterm birth and early onset neonatal infections are estimated to cause approximately 1.4 million neonatal deaths annually (Weston *et al*, 2011). Currently, there is no effective therapy for prevention of *in utero* infections, preterm births, and stillbirths. An important pathogen that causes perinatal and neonatal infections is Group B streptococci (GBS) or *Streptococcus agalactiae*. GBS are β-hemolytic, Gram-positive bacteria that are typically found as recto-vaginal colonizers in healthy adult women (Badri *et al*, 1977; Dillon *et al*, 1982). However, ascending *in utero* GBS infection increases the risk of preterm, premature rupture of membranes (PPROM), fetal injury, and preterm birth (Matorras *et al*, 1989). As ascending infections cannot be treated by intrapartum antibiotic prophylaxis, new strategies are needed to more effectively treat and prevent *in utero* infections and early onset GBS disease. To develop such strategies, it is paramount to gain a better understanding of GBS virulence factors and how they impact the host immune response.

An important virulence determinant of GBS is the toxin known as β-hemolysin/cytolysin (hereafter referred to as the hemolysin). This toxin is responsible for the characteristic zone of β-hemolysis exhibited by GBS, and hemolytic strains are associated with virulence (Liu *et al*, 2004; Fettucciari *et al*, 2006; Costa *et al*, 2012). Also, hyperhemolytic GBS such as those deficient in the two component system CovR/S (due to absence of repression of the hemolysin biosynthesis operon) are significantly more pathogenic, while non-hemolytic GBS are severely attenuated (Sendi *et al*, 2009; Lembo *et al*, 2010). Despite these advances, studies aimed at understanding the mechanism of action of the GBS hemolysin were confounded by difficulties associated with purifying the toxin. Although previously

1 Department of Pediatric Infectious Diseases, University of Washington and Seattle Children's Research Institute, Seattle, WA, USA
2 Department of Global Health, University of Washington, Seattle, WA, USA
3 Department of Physics, University of Washington, Seattle, WA, USA
4 Department of Medicine, Division of Infectious Diseases and Pharmacology, School of Medicine and Lineberger Comprehensive Cancer Center, University of North Carolina at Chapel Hill, Chapel Hill, NC, USA
5 Maternal and Child Health Research Program, Department of Obstetrics and Gynecology, Center for Research on Reproduction and Women's Health, Perelman School of Medicine, University of Pennsylvania, Philadelphia, PA, USA
6 Department of Comparative Medicine, School of Medicine, University of Washington, Seattle, WA, USA
7 Department of Obstetrics and Gynecology, School of Medicine, University of Washington, Seattle, WA, USA
*Corresponding author. Tel: +1 206 884 7336; Fax: +1 206 884 7311; E-mail: lakshmi.rajagopal@seattlechildrens.org
†These authors contributed equally to this work

suggested to be a protein toxin (Marchlewicz & Duncan, 1981; Pritzlaff *et al*, 2001), we recently showed that the molecule responsible for hemolytic activity of GBS is the ornithine rhamnopolyene lipid/ pigment (Whidbey *et al*, 2013) also known as granadaene (Rosa-Fraile *et al*, 2006). With the identification of the GBS hemolysin as a lipid, understanding how the toxin itself contributes to inflammation, cytotoxicity, and preterm birth is critical for development of neutralizing strategies against the lipid toxin. Previous studies that were performed with hemolytic extracts of GBS had contaminating proteins (Marchlewicz & Duncan, 1981); consequently, the exact mechanisms of pigment-induced cytotoxicity are unclear.

Mechanisms that promote ascending GBS infection and the immune responses invoked during this process also remain poorly defined. Studies using pregnant animal models have shown that intrauterine inflammation caused by bacterial infection triggers disruption of placental membranes leading to fetal injury and preterm birth (Elovitz & Mrinalini, 2004; Equils *et al*, 2009; Vanderhoeven *et al*, 2014). A study using intraperitoneal injection of heat-killed GBS showed that a pan-caspase inhibitor was able to delay, but not prevent, preterm birth in a murine model (Equils *et al*, 2009). Although hemolytic GBS strains have been described to activate the NLRP3 inflammasome in murine dendritic cells and macrophages (Costa *et al*, 2012; Gupta *et al*, 2014), whether the hemolytic pigment/lipid toxin is sufficient for inflammasome activation and whether this leads to pyroptosis are not known. Recently, when exogenous GBS RNA was transfected into murine macrophages, the NLRP3 inflammasome was described to associate with GBS RNA, but inflammasome activation required the presence of hemolytic GBS (Gupta *et al*, 2014); however, the relevance of these findings to GBS infection *in vitro* and *in vivo* is unclear. Because NLRP3 activation occurs only in the presence of hemolytic GBS (Costa *et al*, 2012; Gupta *et al*, 2014), we aimed to understand how the purified GBS pigment activates the inflammasome, induces cell death, and establish the consequence on fetal injury.

Here, we demonstrate that the purified GBS pigment/lipid toxin induces membrane permeabilization and the efflux of intracellular potassium which triggers osmotic lysis in red blood cells and NLRP3 inflammasome and caspase 1 activation leading to pyroptosis in macrophages. In a pregnant murine model of intrauterine infection, hyperhemolytic/hyperpigmented GBS strains increased the incidence of preterm birth and *in utero* fetal death (IUFD) in both an NLRP3 inflammasome-dependent and NLRP3 inflammasome-independent manner. Collectively, these findings provide novel insight into how a bacterial lipid toxin/pigment mediates cell death and demonstrates its relevance to bacterial infection and preterm birth.

# Results

## The GBS lipid toxin lyses red blood cells using a colloidal osmotic mechanism

Previous work from our group demonstrated that hyperhemolytic GBS strains penetrate human placenta and can be associated with women in preterm labor (Whidbey *et al*, 2013). We further showed that hemolytic activity of GBS is due to the ornithine rhamnolipid

pigment (see structure in Fig 1A, (Rosa-Fraile *et al*, 2006)) and not due to any protein toxin (Whidbey *et al*, 2013). Despite these findings, the mechanism of how the hemolytic pigment/lipid toxin causes hemolysis, cytolysis, and inflammation-mediated cell death was not known. To understand how the pigment/lipid toxin lyses host cells, we first examined pigment-mediated lysis of human red blood cells (RBCs). We hypothesized that the pigment may lyse RBC either by the mechanism of direct lysis where the lipid itself dissolves the membrane as observed with detergents or by the mechanism of colloidal osmotic lysis where the lipid forms pores or causes membrane perturbations and lysis occurs via osmotic pressure. To determine how the GBS pigment induces cell lysis, we first measured the kinetics of both $K^+$ and hemoglobin (Hb) release from RBC treated with 400 nM pigment. As a control, an equal amount of extract from a non-hemolytic strain of GBS ($\Delta cylE$) was included. The results shown in Fig 1B indicate that while the pigment induced the release of both $K^+$ and Hb from RBC, efflux of the smaller $K^+$ ion was faster than efflux of the larger Hb, as measured by time to 50% release (Fig 1B; 4.8 min versus 8.4 min; $P < 0.0001$, extra sum-of-squares F test). These results suggest that the pigment induces membrane permeabilization that allows $K^+$ ions to efflux, followed by the release of Hb. The slight lag in release of Hb versus $K^+$ suggests a colloidal osmotic mechanism of lysis by the GBS pigment, rather than rapid dissolution of the membrane by direct lysis wherein no lag is expected between $K^+$ and Hb. A similar lag between $K^+$ and Hb release was observed during hemolysis mediated by *Staphylococcus aureus* α-toxin (Supplementary Fig S1A), whereas 100% release of both $K^+$ and Hb occurred instantly with direct lysis mediated by Triton X-100 (Supplementary Fig S1B).

We also performed protection assays with osmoprotectants of various sizes ranging from a hydrodynamic radius of 0.40 nm (PEG200) to 1.6 nm (PEG3000). To this end, human RBCs were pretreated with the GBS pigment for 2 min and the RBCs were pelleted to remove any unincorporated pigment. Pigment-treated RBCs were then resuspended in PBS or PBS containing 30 mM osmoprotectant, and hemolytic activity was measured (for details, see Materials and Methods). The results shown in Fig 1C indicate that smaller osmoprotectants such as PEG200 and PEG400 did not protect RBC from pigment-mediated hemolysis, whereas complete protection from hemolysis was observed in the presence of the larger osmoprotectants such as PEG1500 and PEG3000. In comparison, minimal protection was observed with SDS, which causes direct and instant lysis of RBC (Supplementary Fig S2).

## The GBS hemolytic lipid induces membrane permeabilization of artificial lipid bilayers

To determine if membrane permeability observed with the GBS pigment requires the active cellular response of host cells, we tested the ability of the pigment to disrupt artificial lipid bilayers using model black lipid membranes (BLMs). BLMs mimic membrane lipid bilayers but lack the active cellular responses of host cells. To test our hypothesis, BLM composed of 1,2-diphytanoyl-sn-glycero-3-phosphocholine (DPhPC) was established across an aperture separating two chambers of a U-tube as previously described Butler *et al* (2008). A voltage was applied, and current was measured between the two chambers; an increase in current corresponds to a compromise in

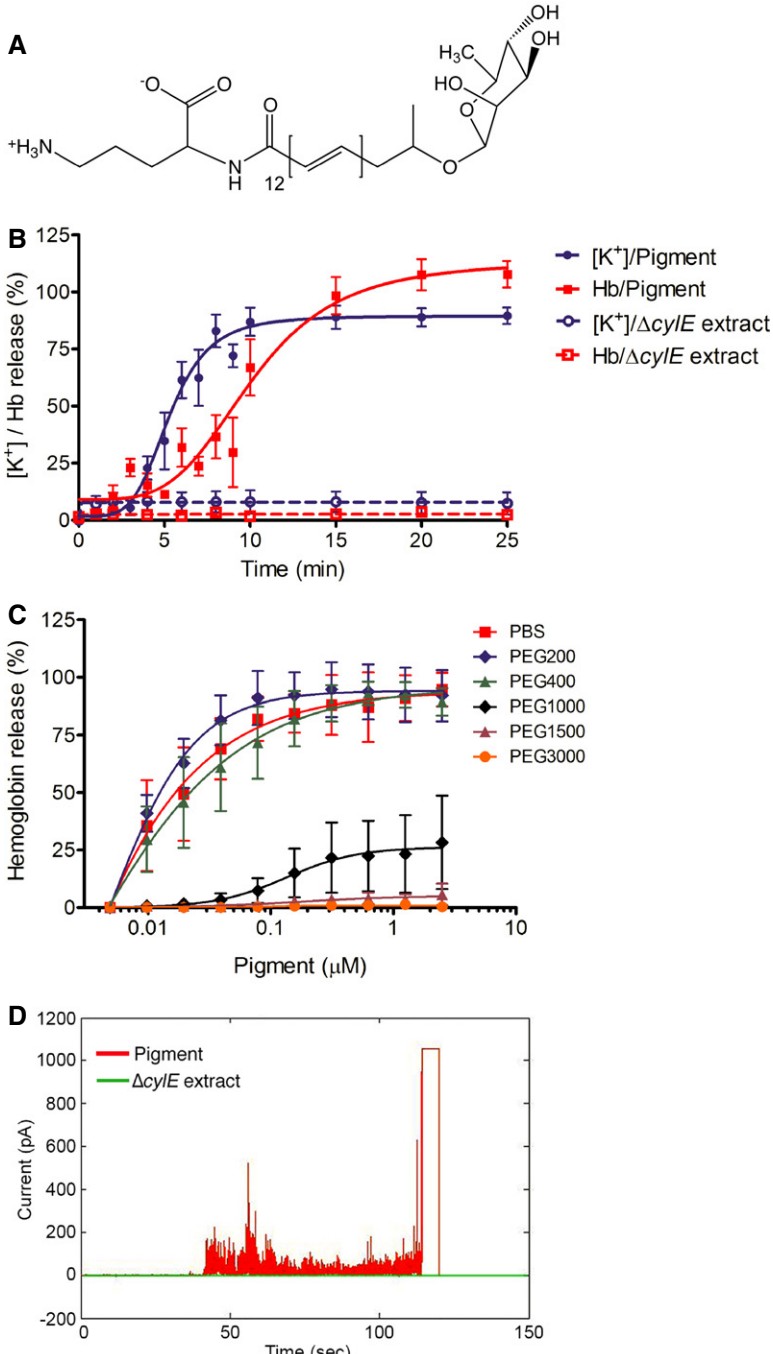

**Figure 1.  Colloidal osmotic lysis and membrane permeabilization caused by the GBS pigment/lipid toxin.**

A   The GBS pigment also known as granadaene is an ornithine rhamnopolyene (Rosa-Fraile et al, 2006).

B   Human red blood cells (RBCs) were incubated with 400 nM pigment or control $\Delta cylE$ extract, and kinetics of $K^+$ and Hb release was monitored. Data shown are the average and SEM of six independent experiments. The time to 50% $K^+$ and 50% Hb release with pigment was 4.8 min and 8.4 min, respectively; $n = 6$, $P < 0.0001$, extra sum-of-squares F test.

C   Role of osmoprotectants in pigment-treated RBC. Human RBCs were pre-incubated with GBS pigment for 2 min at RT, centrifuged, and resuspended in the presence and absence of 30 mM osmoprotectant with hydrodynamic radius of 0.40 nm (PEG200), 0.56 nm (PEG400), 0.89 nm (PEG1000), 1.1 nm (PEG1500), or 1.6 nm (PEG3000), respectively. Release of Hb was measured after 1 h of incubation at 37°C. Data shown are the average and SEM of three independent experiments.

D   Characteristics of membrane permeabilization by the GBS pigment in artificial lipid bilayers. Lipid bilayers were generated using diphytanoylphosphatidylcholine (DPhPC) and treated with either 2 μM pigment or an equivalent amount of the control $\Delta cylE$ extract. In the pigment-treated sample, channel conductance indicating disruption of the membrane is seen within 45 s. Erratic and non-discrete fluctuations in current are observed, suggesting the formation of multiple, small membrane defects. The bilayer eventually breaks at 120 s. In lipid bilayers treated with the control $\Delta cylE$ extract, the mean ionic current trace remains constant at 0 pA, showing no membrane disruption. Data shown are representative of three independent experiments.

membrane integrity. We observed that treatment of BLMs with pigment resulted in an increase in measured current, while treatment with the control ΔcylE extract resulted in no change (data from 2 μM are shown in Fig 1D and from 75 nM are shown in Supplementary Fig S3). Interestingly, jumps were erratic in both frequency and magnitude. The initial compromised bilayer area was in the order of 1 nm². Subsequently, the size of the compromise in the bilayer fluctuated frequently before the bilayer finally ruptured at either 120 s (Fig 1D) or 345 s (Supplementary Fig S3), depending on pigment concentration. The increase in conductance observed in the lipid bilayers due to the GBS pigment is not canonical with either the formation of discrete protein pores that are usually marked by well-defined jumps in membrane conductance or detergent-mediated bilayer solubilization, which is marked by a rapid increase in conductance followed by spontaneous disappearance of the entire membrane (Supplementary Fig S3). The changes in conductance fluctuations are consistent with the pigment intercalating into and perforating the bilayer; this appears to be a dynamic process in which channel conductance appears and disappears. Taken together, these data demonstrate that the membrane permeability observed in pigment-treated cells occurs independently of a cellular response and is unique in its mode of action by neither conforming to a typical pore-forming protein toxin nor inducing instant lysis as observed with detergents.

### Purified GBS lipid toxin/pigment is sufficient for induction of IL-1β release and cytolysis

To understand how the hemolytic pigment triggers host cell death, we examined the pro-inflammatory and cytotoxic properties elicited by the purified hemolytic pigment. In these experiments, we also included GBS strains with differences in hemolytic activity such as GBS WT A909, isogenic hyperhemolytic/hyperpigmented strain ΔcovR (lacking the two component regulator CovR/S that represses biosynthesis of the hemolytic pigment), and non-hemolytic ΔcylE, ΔcovRΔcylE strains that were derived from WT and ΔcovR but lack the cylE gene important for pigment biosynthesis (for details, see (Whidbey et al, 2013)). Given that macrophages are important for defense against GBS infections, we utilized macrophages derived from M-CSF-treated human peripheral blood mononuclear cells (PBMC) as well as differentiated THP-1 cells as models of human macrophage-like cells. Human PBMC and THP-1-derived macrophages were treated with GBS WT, hyperhemolytic ΔcovR, and non-hemolytic ΔcylE and ΔcovRΔcylE at a multiplicity of infection (MOI) of 1 for 4 h, and cytotoxicity was measured by LDH release. The results shown in Figs 2A and 3A show that LDH release indicative of cell death was significantly higher in macrophages treated with hyperhemolytic GBS when compared to non-hemolytic strains. We also observed that hyperhemolytic GBS induced significantly more IL-1β release when compared to the isogenic non-hemolytic strains (Figs 2B and 3B).

To determine the importance of the pigment/lipid toxin in the pro-inflammatory and cytotoxic nature of GBS, the human PBMC- and THP-1-derived macrophages were incubated with various concentrations of purified pigment or control ΔcylE extract for 4 h, and cytotoxicity was measured as the loss of metabolic activity as measured by a redox dye, alamar blue. Cytotoxicity due to GBS pigment was dose dependent and 50% cell death was observed at

approximately 1–2 μM (Figs 2C and 3C), which is noticeably higher than the EC$_{50}$ for RBC (< 0.1 μM, Fig 1C). This observation is likely due to the membrane turnover that occurs in macrophages and not RBC, and has also been observed with other bacterial exotoxins (Keyel et al, 2011).

To identify the pathways activated by the purified GBS pigment/ lipid toxin, we measured cytokine levels in the supernatants of pigment-treated macrophages. Interestingly, levels of IL-1β were significantly increased in pigment-treated PBMC and THP-1 cells compared to control ΔcylE extract-treated cells (Figs 2D and 3D). Consistent with the above observations, IL-18 levels were also increased in pigment-treated THP-1 macrophages, but other pro-inflammatory cytokines such as TNFα, IL-6, and IFN-γ were not significantly increased (Supplementary Fig S4). Taken together, these data show that the purified hemolytic GBS pigment is pro-inflammatory and cytotoxic.

### Pigment-induced cytotoxicity and immune response are NLRP3 inflammasome dependent

The increase in secretion of IL-1β and IL-18 observed in human macrophages treated with GBS pigment suggests that the pigment can trigger activation of the inflammasome. The inflammasome is a cytosolic complex, which mediates cleavage of pro-caspase 1 to active caspase 1, which in turn cleaves pro-IL-1β and pro-IL-18 into their active forms. One major inflammasome comprises the NLR (nucleotide binding, leucine-rich repeat containing) protein known as NLRP3, which associates with the adaptor ASC (apoptosis-associated speck-like protein containing the caspase recruitment domain, CARD) (Taxman et al, 2010). To determine if the GBS pigment/lipid toxin activates NLRP3, we exposed previously characterized THP-1 human macrophage cell lines that were constitutively knocked down for expression of NLRP3 or the adapter protein ASC ((Willingham et al, 2007), Supplementary Fig S5) to GBS strains (WT, isogenic ΔcovR, ΔcylE, ΔcovRΔcylE) or purified pigment. As controls, THP-1 macrophages transfected with empty vector or a shRNA of scrambled ASC sequence were included. The results shown in Fig 4A indicate that GBS induced significant cell death in macrophages containing the NLRP3 inflammasome in a hemolysin-dependent manner. Notably, cell death was significantly decreased in macrophage cell lines knocked down for expression of NLRP3 or ASC (Fig 4A). Similarly, hemolytic and hyperhemolytic GBS strains induced increased IL-1β secretion in macrophages in an NLRP3 inflammasome-dependent manner (Fig 4B). Consistent with these observations, we observed that increasing concentrations of the purified GBS hemolytic pigment induced cell death and IL-1β secretion in an NLRP3 inflammasome-dependent manner (Fig 4C and D). These results demonstrate that in macrophages, the GBS hemolytic pigment primarily induces an NLRP3 inflammasome-dependent programmed cell death. The cell death observed in NLRP3-deficient macrophages with hyperpigmented GBSΔcovR (25–40%, see THP-1/shASC, THP-1/shNLRP3 in Fig 4A) could not be prevented by the addition of osmoprotectants such as PEG1500 or by the caspase 3/7 inhibitor Z-DEVD-FMK (Supplementary Fig S6). Also, levels of IL-1β released by THP-1/shASC and THP-1/shNLRP3 cells exposed to GBSΔcovR were not significantly different from cells exposed to GBS WT, ΔcylE, or ΔcovRΔcylE (Fig 4B). Based on these observations, we predict that the residual cell death observed in inflammasome-deficient cells due

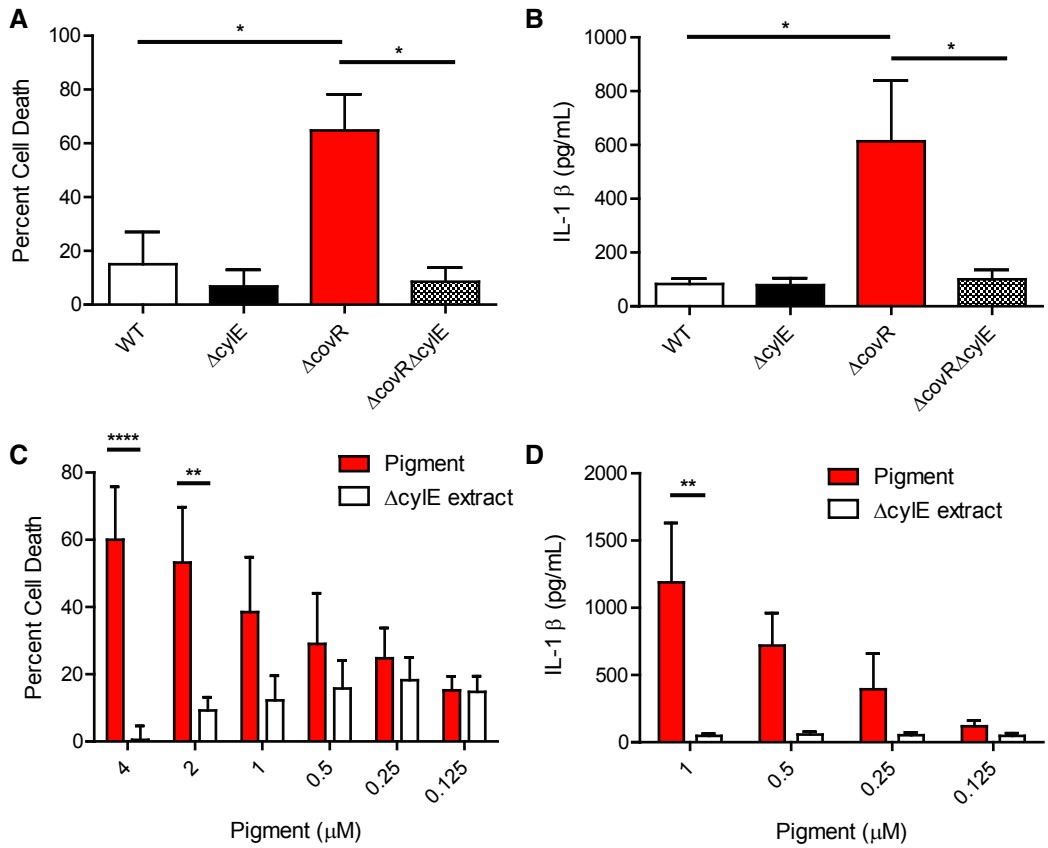

**Figure 2.  The GBS pigment toxin/lipid toxin is pro-inflammatory and cytotoxic to primary human macrophages.**

A, B  PBMC-derived macrophages were treated with GBS WT, ΔcylE, ΔcovR, or ΔcovRΔcylE at an MOI of 1 and incubated for 4 h. Cytotoxicity was measured by LDH release (A), and IL1β release in supernatants was measured by ELISA (B).

C, D  PBMC-derived macrophages primed with 100 ng/ml LPS for 3 h were incubated with various concentrations of GBS pigment or control ΔcylE extract for 4 h. Cytotoxicity was measured by alamar blue assay (C), and IL1β release from pigment- or ΔcylE extract-treated cells was measured by Luminex assay (D).

Data information: Data shown are the average of four independent experiments performed in triplicate, error bars ± SEM. Significance was determined using Bonferroni's multiple comparison test following ANOVA. (A) $n = 4$, *$P = 0.021$ for WT versus ΔcovR; *$P = 0.01$ for ΔcovR versus ΔcovRΔcylE. (B) $n = 4$, *$P = 0.031$ for WT versus ΔcovR; *$P = 0.036$ for ΔcovR versus ΔcovRΔcylE. (C) $n = 4$, ****$P < 0.0001$, **$P = 0.002$. (D) $n = 4$, **$P = 0.005$.

to hyperpigmented GBS can be attributed to an inflammasome- and caspase 3/7-independent pathway, as suggested previously with rat neuronal cells (Reiss et al, 2011).

### The GBS pigment induces membrane permeabilization and loss of intracellular potassium independent of the NLRP3 inflammasome

A known activator of the NLRP3 inflammasome is the efflux of intracellular potassium that occurs upon membrane permeabilization (Munoz-Planillo et al, 2013). Given that the GBS pigment is able to induce membrane disruptions in RBC and even in artificial lipid bilayers (Fig 1C and D), we hypothesized that intercalation of the GBS pigment into host cells such as human macrophages should trigger membrane permeabilization and the efflux of intracellular potassium, irrespective of the presence or absence of the inflammasome. To test this hypothesis, we measured membrane disruption and quantified intracellular potassium levels in NLRP3-proficient and NLRP3-deficient macrophages that were treated with the GBS pigment. To examine membrane permeabilization, we exposed

NLRP3-proficient and NLRP3-deficient macrophages to GBS pigment (1 μM) or controls for 20 min and measured uptake of a membrane impermeable dye, propidium iodide (see Materials and Methods for details). The results shown in Fig 5A and B indicate that increased fluorescence is seen in both NLRP3-proficient and NLRP3-deficient macrophages treated with GBS pigment (1 μM) when compared to controls (ΔcylE extract). These data indicate that the GBS pigment induces membrane permeability in host cells independent of the inflammasome. To further confirm this, we utilized ion-coupled plasma atomic emission spectroscopy (ICP-AES) as described (Munoz-Planillo et al, 2013) to measure levels of intracellular potassium in NLRP3-proficient and NLRP3-deficient macrophages that were exposed to GBS pigment or to controls over time (0, 30, 60, 120, 180, and 240 min). These results demonstrate that intracellular potassium levels dramatically decreased within 30 min in NLRP3-proficient macrophages (THP-1/scrambled, Fig 5C) and in NLRP3-deficient macrophages (THP-1/shNLRP3, Fig 5D) that were exposed to the GBS pigment (1 μM), demonstrating that both pigment-mediated membrane disruption and potassium efflux occur independently of NLRP3 inflammasome activation and even cell death. Notably,

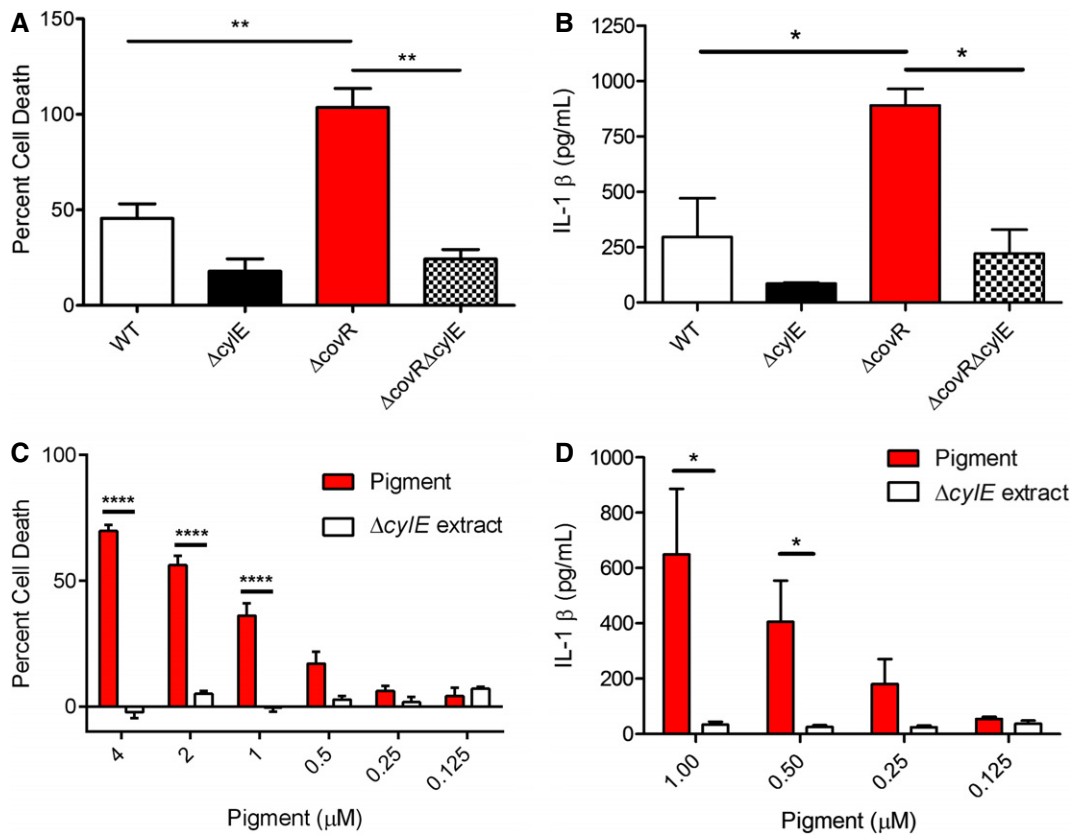

**Figure 3. The GBS pigment toxin/lipid toxin is pro-inflammatory and cytotoxic to immortalized THP-1 monocyte-derived macrophages.**

A, B   WT THP-1 macrophages were treated with GBS WT, Δ*cylE*, Δ*covR*, or Δ*covR*Δ*cylE* at an MOI of 1 and incubated for 4 h. Cytotoxicity was measured by LDH release (A), and IL1β release in supernatants was measured by Luminex assay (B).

C, D   WT THP-1 macrophages were incubated with various concentrations of GBS pigment or control Δ*cylE* extract for 4 h. Cytotoxicity was measured by alamar blue assay (C), and IL1β release from pigment- or Δ*cylE* extract-treated cells was measured by Luminex assay (D).

Data information: Data from three independent experiments performed in triplicate are shown, error bars ± SEM. Significance was determined using Bonferroni's multiple comparison test following ANOVA. (A) $n = 3$, **$P = 0.0080$ for WT versus Δ*covR*; **$P = 0.0016$ for Δ*covR* versus Δ*covR*Δ*cylE*. (B) $n = 3$, *$P = 0.03$ for WT versus Δ*covR*; *$P = 0.02$ for Δ*covR* versus Δ*covR*Δ*cylE*. (C) $n = 3$, ****$P < 0.0001$. (D) $n = 3$, *$P = 0.01$.

NLRP3-deficient macrophages appear to be able to recover from the initial $K^+$ efflux (see 120, 180, and 240 min in Fig 5D). In contrast, cells proficient for NLRP3 undergo cell death and do not significantly recover (Fig 5C). To determine if $K^+$ efflux was important for pigment-induced cell death, we performed cytolysis assays of THP-1 cells exposed to the GBS pigment in the presence of excess extracellular potassium (final concentration 50 mM) compared to culture media containing 5 mM potassium. The results shown in Fig 5E indicate that addition of extracellular potassium partially protected the cells from GBS pigment-mediated cytolysis, further supporting a role of $K^+$ efflux and NLRP3 activation in cytolysis.

**The GBS pigment induces caspase 1-dependent pyroptosis**

Activation of the NLRP3 inflammasome by $K^+$ efflux leads to activation of caspase 1 (Petrilli *et al*, 2007; Jin & Flavell, 2010; Davis *et al*, 2011; Munoz-Planillo *et al*, 2013). To determine if the GBS pigment-mediated $K^+$ efflux induced caspase 1 activation, we utilized a membrane permeable, caspase 1-specific FLICA reagent (fluorochrome-labeled inhibitors of caspases, 660-YVAD-FMK;

Immunochemistry Technologies) that binds to active caspase 1. We observed that in macrophages expressing NLRP3, pigment treatment resulted in more FLICA+ cells when compared to pigment treatment of macrophages that were knocked down for NLRP3 expression (Fig 6A). Collectively, these results show that the GBS pigment triggers potassium efflux and NLRP3 inflammasome-mediated caspase 1 activation. Active caspase 1 can trigger the pro-inflammatory programmed cell death known as pyroptosis, a type of cell death that has been increasingly associated with bacterial pathogenesis (LaRock & Cookson, 2013). To determine if the GBS pigment induces cell death by pyroptosis, we utilized the caspase 1-specific inhibitor Z-YVAD-FMK. We observed that when THP-1 macrophages were pre-treated with the caspase 1-specific inhibitor, they were significantly more resistant to pigment-mediated cell death (Fig 6B), whereas protection from pigment-mediated cell death was not observed with the caspase 3/7 inhibitor Z-DEVD-FMK or vehicle only (Fig 6B). Also, increasing the concentration of the caspase 1-specific inhibitor Z-YVAD-FMK provided dose-dependent protection from pigment-mediated cell death and complete protection was seen at 200 μM Z-YVAD-FMK (Supplementary Fig S7). The requirement

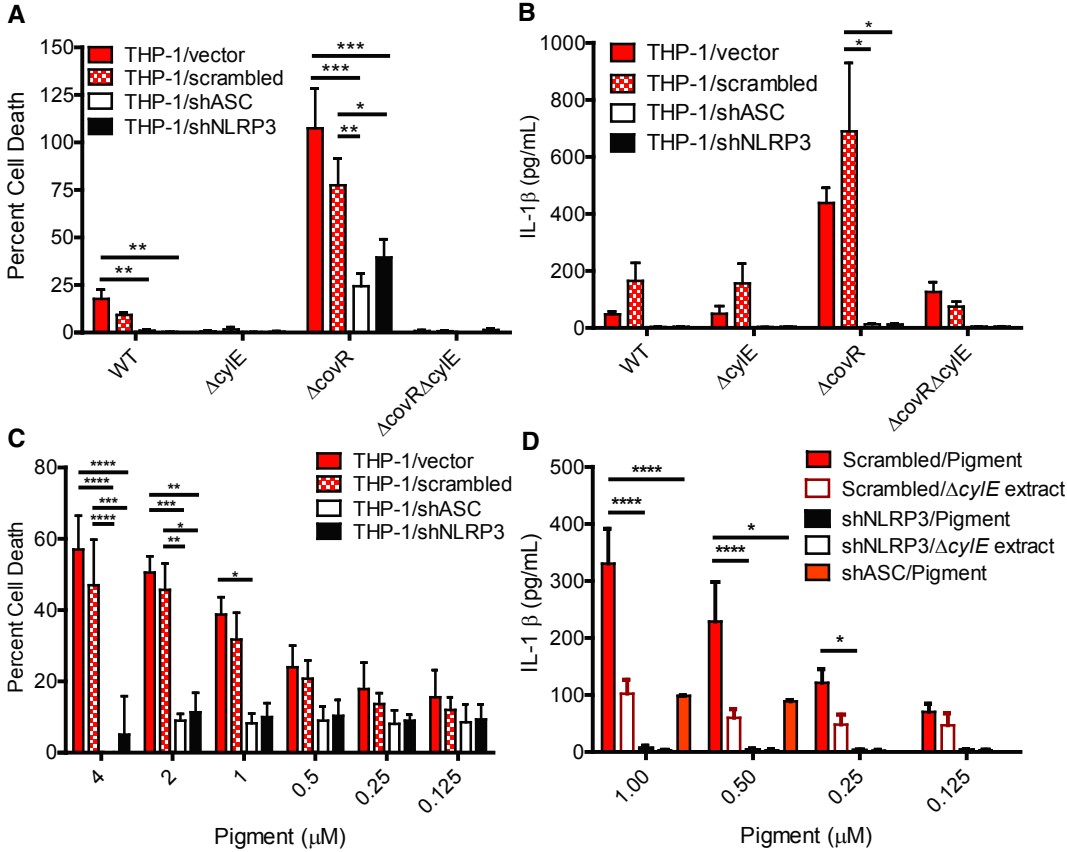

**Figure 4.** The GBS pigment induces NLRP3 inflammasome-dependent cell death in human macrophages.

A, B  THP-1 macrophages transfected with empty vector, scrambled control, shASC, or shNLRP3 were treated with GBS WT, ΔcylE, ΔcovR, or ΔcovRΔcylE at an MOI of 1 and incubated for 4 h. Cytotoxicity was measured by LDH release (A), and IL1β release in supernatants was measured by Luminex assay (B). Data from three independent experiments performed in triplicate are shown.

C, D  The shRNA THP-1 macrophages were incubated with various concentrations of GBS pigment or ΔcylE extract for 4 h. Cytotoxicity was measured by alamar blue assay (C), and IL1β release in supernatants was measured by Luminex assay (D). Pigment-mediated cytotoxicity is dependent on the NLRP3 inflammasome components, suggesting that pigment is inducing a programmed cell death. Data shown are the average of at least three independent experiments performed in triplicate.

Data information: Data were analyzed using Bonferroni's multiple comparison test following ANOVA, error bars ± SEM. (A) $n = 3$, for WT: **$P = 0.001$ (vector versus shNLRP3), **$P = 0.0016$ (vector versus shASC); for ΔcovR: ***$P = 0.0008$ (vector versus shNLRP3), ***$P = 0.0001$ (vector versus shASC), *$P = 0.047$ (scrambled versus shNLRP3), **$P = 0.0054$ (scrambled versus shASC). Data obtained from THP-1/vector was not significantly different from THP-1/scrambled, $P = 0.18$. (B) $n = 3$, *$P = 0.025$. Data obtained from THP-1/vector were not significantly different from THP-1/scrambled, $P = 0.89$. (C) $n = 3$, for 4 μM pigment: **$P = 0.0063$ (vector versus shNLRP3), **$P = 0.0035$ (vector versus shASC), *$P = 0.01$ (scrambled versus shNLRP3), **$P = 0.009$ (scrambled versus shASC); for 2 μM pigment: **$P = 0.0088$ (vector versus shNLRP3), **$P = 0.0066$ (vector versus shASC), *$P = 0.01$ (scrambled versus shNLRP3 or shASC); for 1 μM pigment: *$P = 0.02$ (vector versus shNLRP3), *$P = 0.01$ (vector versus shASC). Data obtained from THP-1/vector were not significantly different from THP-1/scrambled, $P = 0.99$. (D) $n = 3$, ****$P < 0.0001$, *$P = 0.015$.

of $K^+$ efflux, NLRP3, ASC, caspase 1, and the ensuing pro-inflammatory response demonstrates that GBS utilizes a hemolytic pigment/lipid toxin to induce pyroptosis.

**The GBS pigment causes fetal injury by NLRP3 inflammasome-dependent and NLRP3 inflammasome-independent mechanisms**

Previous work from our group demonstrated that hyperhemolytic GBS strains with mutations in CovR/S were more proficient in penetration of human placenta and were also isolated from chorioamniotic membranes and amniotic fluid from women in preterm labor (Whidbey *et al*, 2013). To determine if hyperhemolytic GBS strains induce fetal injury and preterm birth, we adapted a murine model of *E. coli*-induced *in utero* infection (Elovitz *et al*, 2003) to GBS. We chose the intrauterine model of inoculation rather

than a vaginal model of inoculation to minimize discrepancies that can be attributed to differences in mouse vaginal persistence (Patras *et al*, 2013). To this end, on day E14.5 of pregnancy, either WT GBS, the hyperpigmented ΔcovR, or the non-hemolytic ΔcovRΔcylE was infused into the right horn of the uterus between the first (P1) and second (P2) fetal sacs most proximal to the cervix as described (Hirsch *et al*, 1995; Elovitz *et al*, 2003) (also see Fig 7A). Mice were monitored for 72 h for signs of preterm birth (i.e., at least 1 pup in cage). At either 72 h or at the onset of delivery (whichever occurred first), fetal survival rates were determined and tissue was collected to assess bacterial load. Preterm birth (i.e., at least 1 pup in cage) was observed in three of six mice infected with ΔcovR, in one of six infected with WT GBS, and in none of the six mice infected with the non-hemolytic ΔcovRΔcylE strain (preterm delivery rates were 50, 16, and 0%, respectively). We also observed significantly higher

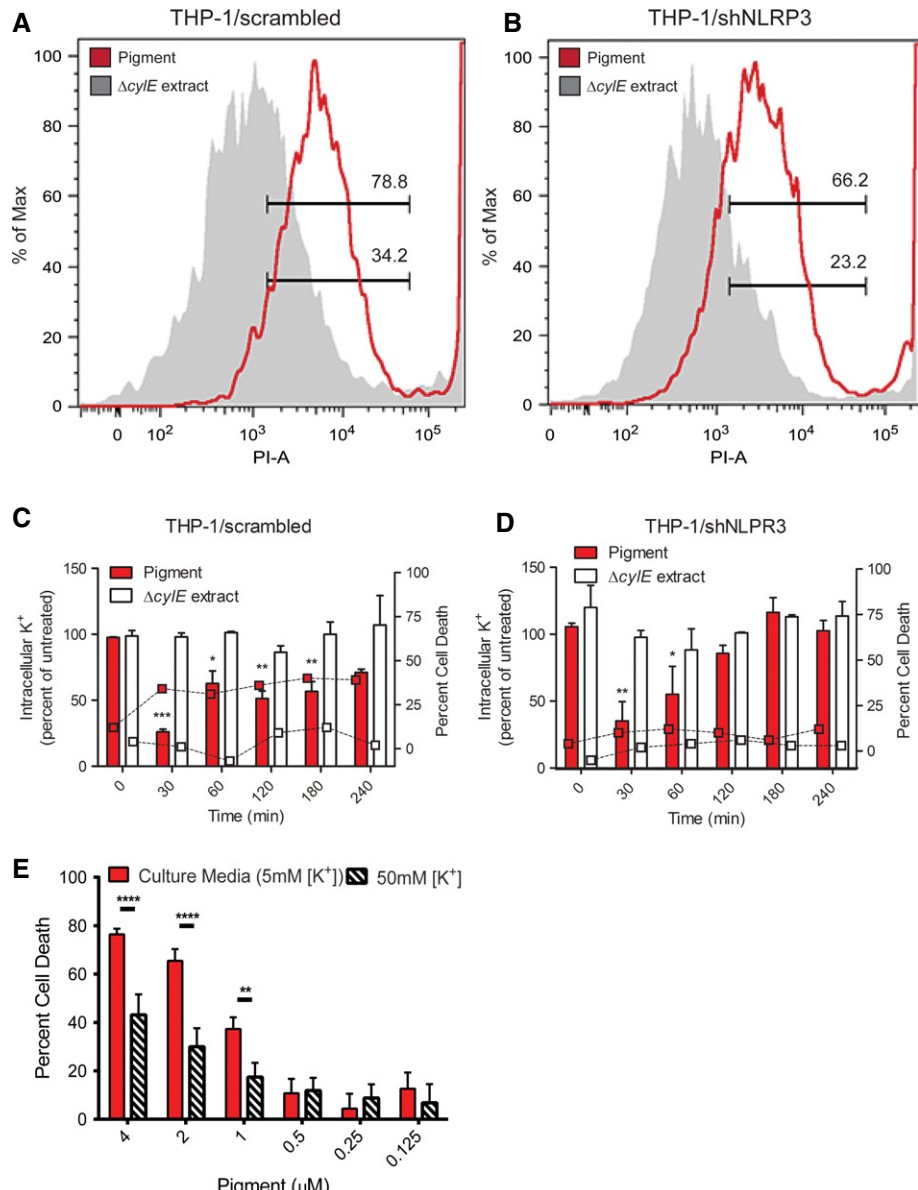

**Figure 5.  GBS pigment induces membrane permeabilization and K⁺ efflux independently of the NLRP3 inflammasome.**

A, B   THP-1 macrophages proficient for NLRP3 (transfected with scrambled control, A) or deficient for NLRP3 (shNLRP3, B) were treated with 1 μM pigment or ΔcylE
extract for 20 min, and propidium iodide (PI) was added during the final 10 min. PI uptake was measured by flow cytometry, and data shown are representative of
two independent experiments.

C, D   Intracellular potassium concentration was measured by ICP-AES. THP-1 macrophages transfected with the scrambled control (C) or shNLRP3 (D) were treated with
GBS pigment (1 μM) or an equivalent amount of the ΔcylE extract. At various time points, cells were lysed and intracellular [K⁺] was measured relative to
untreated cells (see bars and left y-axis), and percent cell death was quantified by alamar blue (see squares, dotted connecting lines and right y-axis). Both NLRP3-
proficient and NLRP3-deficient macrophages initially lose intracellular K⁺ due to GBS pigment (compare t = 0 min to t = 30 min), but the NLRP3-deficient cells
(shNLRP3) are able to recover, while the scrambled control do not, demonstrating that initial K⁺ loss occurs independently of NLRP3. Data are average of three
independent experiments performed with independent pigment preparations in triplicate and were analyzed using Dunnett's multiple comparison test following
ANOVA; all data were compared to control at t = 0, error bars ± SEM. (C) n = 3, ***P = 0.0002, *P = 0.019, **P = 0.0032 for 120 min, **P = 0.0072 for 180 min.
(D) n = 3, **P = 0.0043, *P = 0.031. (E) WT THP-1 macrophages were incubated with pigment in media containing either 5 mM or 50 mM potassium chloride, and
cytotoxicity was measured by alamar blue assay. The addition of potassium chloride is able to protect the macrophages from cytolysis, demonstrating that K⁺
efflux is essential for this process. Data shown are three independent experiments performed in triplicate and were analyzed using Bonferroni's multiple
comparison test following ANOVA, error bars ± SEM (n = 3, ****P < 0.0001, **P = 0.0036).

fetal death in mice infected with hemolytic GBS, that is, WT or hyperpigmented ΔcovR compared to the non-hemolytic ΔcovRΔcylE strain (Fig 7B). Bacteria were recovered from all fetuses present in

the uterus, and there was no significant difference in bacterial load between fetuses of mice infected with the three GBS strains (data not shown). H&E staining of infected uterine tissue showed that

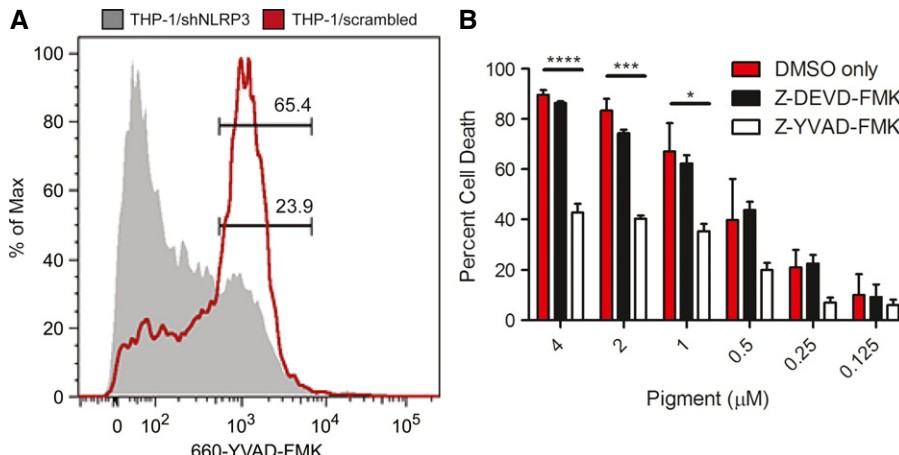

**Figure 6. The GBS hemolytic pigment/lipid toxin induces caspase 1 activation and pyroptosis.**

A  THP-1 macrophages were treated with GBS pigment, and caspase 1 activation was measured by flow cytometry using a FLICA reagent. Pigment treatment of the scrambled shRNA control cell line induces more caspase 1 activation compared to the shNLRP3 cell line, demonstrating that the pigment activates caspase 1 exclusively through the NLRP3 inflammasome. Results are representative of three independent experiments.

B  WT THP-1 macrophages were treated with the caspase 1 inhibitor Z-YVAD-FMK, the caspase 3/7 inhibitor Z-DEVD-FMK, or DMSO only prior to treatment with the GBS pigment. YVAD is able to significantly decrease cytotoxicity in cells treated with GBS pigment, demonstrating that caspase 1 is required for GBS pigment-mediated cell death, characteristic of pyroptosis, while DEVD had no effect. Data are average of three independent experiments, error bars $\pm$ SEM. Significance was determined using Bonferroni's multiple comparison test following ANOVA ($n = 3$, ****$P < 0.0001$, ***$P = 0.0002$, *$P = 0.014$).

while only a few mononuclear cells are present in the $\Delta covR\Delta cylE$ sample, increased presence of inflammatory cells and necrotic debris is seen in the $\Delta covR$ sample (Fig 7C). The frequency of fetal death between hemolytic strains of GBS such as WT and $\Delta covR$ was not significantly different and may likely be due to decreased repression of hemolysin/pigment biosynthetic genes by CovR/S *in utero* as suggested previously (Santi *et al*, 2009; Sitkiewicz *et al*, 2009).

To determine if hemolysis and/or activation of the NLRP3 inflammasome is important for fetal injury and preterm birth caused by hyperhemolytic GBS strains, we utilized pregnant homozygous NLRP3 knockout (NLRP3KO) mice (Brydges *et al*, 2009; Kovarova *et al*, 2012) in the *in utero* model of infection described above. We compared the frequency of preterm birth and fetal death in pregnant WT C57BL6 and NLRP3KO mice that were infected *in utero* with either hyperhemolytic GBS$\Delta covR$ or isogenic non-hemolytic GBS$\Delta covR\Delta cylE$. Notably, preterm delivery was observed in three of six WT C57BL6 mice infected with $\Delta covR$ and not in any other groups, that is, WT C57BL6 mice infected with $\Delta covR\Delta cylE$ or NLRP3KO mice infected with either $\Delta covR$ or $\Delta covR\Delta cylE$. The results shown in Fig 7D indicate that GBS infection-mediated fetal death can be associated with production of the hemolytic pigment (compare $\Delta covR$ and $\Delta covR\Delta cylE$ in WT C57BL6 mice) and the presence of the NLRP3 inflammasome (compare $\Delta covR$ in WT C57BL6 and NLRP3KO mice). Fetal death was also significantly higher in NLRP3KO mice infected with $\Delta covR$ compared to NLRP3KO mice infected with $\Delta covR\Delta cylE$; these results indicate that the hemolytic/membrane-disrupting nature of the pigment (without NLRP3 inflammasome activation) is also likely to contribute to fetal injury. Both hemolytic and non-hemolytic GBS were able to penetrate all fetuses located throughout the uterus in both WT and NLRP3KO mice (Fig 7E), indicating that decreased fetal death observed in the inflammasome-deficient

mice or due to non-hemolytic/non-pigmented GBS strains cannot be attributed to decreased bacterial dissemination. Of note, fetal death observed in NLRP3KO mice due to non-hemolytic GBS is not significantly different when compared to WT C57BL6 mice ($P > 0.3$, Fig 7D) or CD-1 mice ($P = 1$ compare Fig 7D to B). IL-1$\beta$ levels were significantly higher in WT C57BL6 mice infected with $\Delta covR$ compared to NLRP3KO mice infected with $\Delta covR$ or WT mice infected with $\Delta covR\Delta cylE$ (Fig 7F). As IL-1$\beta$ levels were not significantly different between NLRP3KO mice infected with $\Delta covR$ or $\Delta covR\Delta cylE$, these results suggest that the GBS pigment primarily activates the NLRP3 inflammasome *in vivo*. IL-1$\beta$ levels in WT C57BL6 mice infected with $\Delta covR\Delta cylE$ were higher than NLRP3KO mice infected with $\Delta covR\Delta cylE$, suggesting that additional factors besides the pigment also activate the NLRP3 inflammasome, but interestingly, this did not correlate to increased fetal death. Collectively, our results indicate that production of the hemolytic pigment contributes to GBS infection-associated fetal injury in both an NLRP3 inflammasome-dependent and NLRP3 inflammasome-independent manner and places the hemolytic pigment as a critical component of GBS fetal injury.

## Discussion

This study demonstrates how the GBS hemolytic pigment/lipid toxin causes cell death, triggers inflammation, and promotes fetal injury. Although the hemolytic nature of GBS was described almost a century ago (Brown, 1920; Rosa-Fraile *et al*, 2014) and is associated with virulence of the pathogen (Pritzlaff *et al*, 2001; Doran *et al*, 2003; Liu *et al*, 2004; Hensler *et al*, 2005; Lembo *et al*, 2010), mechanistic insight into this potent virulence factor was lacking. We recently showed that the molecular basis of hemolytic activity in GBS is the pigment/lipid toxin (Whidbey

**Figure 7.  The GBS pigment causes fetal injury by NLRP3 inflammasome-dependent and NLRP3 inflammasome-independent mechanisms.**
Female pregnant wild-type (CD-1, C57BL6) or NLRP3-deficient mice were injected *in utero* with $10^{6–7}$ CFU of GBS WT, ΔcovR, or ΔcovRΔcylE and monitored for preterm birth. Surgery and GBS inoculation for each pregnant mouse were performed independently. Data shown are representative of experiments with 6 animals per group for each GBS strain and two animals were used for saline controls.

A    Scheme of pup numbering *in utero* and injection site between fetuses P1 and P2 is shown.
B    *In utero* fetal death in wild-type CD-1 mice due to infection with GBS WT, hyperhemolytic ΔcovR, and non-hemolytic ΔcovRΔcylE. Fetal death is represented by the number of dead fetuses/total number of fetuses obtained from six pregnant mice per group. '*n*' indicates total number of pups (both live and dead); ****$P < 0.0001$, Fisher's exact test.
C    H&E staining of uterine tissue. Open arrow indicates the presence of few mononuclear cells, whereas filled arrows indicate increased infiltration of inflammatory cells and necrotic debris.
D    Fetal death due to infection with hyperhemolytic GBS ΔcovR and non-hemolytic ΔcovRΔcylE in WT C57BL6 and NLRP3 inflammasome-deficient mice; fetal death is represented by the number of dead fetuses/total number of fetuses obtained from 6 pregnant mice per group. '*n*' indicates total number of pups (both live and dead); *$P = 0.011$, **$P = 0.0015$, ****$P < 0.0001$, Fisher's exact test. Fetal death due to ΔcovRΔcylE in WT C57BL6 and NLRP3KO mice was not significant and is indicated as NS; $P = 0.31$, Fisher's exact test.
E    Bacterial burden in fetal pups and uterine horns from mice infected with the various GBS strains (*n* = 6/pup; of note, pups that were delivered preterm were excluded from CFU enumeration. Scheme of pup numbering is shown in (A). RUH and LUH indicate right uterine horn and left uterine horn, respectively. CFUs are not significantly different between any of the groups (ANOVA, $P = 0.6$, error bars ± SEM).
F    IL-1β levels in GBS-infected tissues (placenta and fetus, *n* = 6/group) was measured by Luminex assay (*$P = 0.025$, ***$P = 0.0002$, ****$P < 0.0001$, Bonferroni's multiple comparison test following ANOVA. IL-1β levels was not significantly different in NLRP3KO mice infected with ΔcovR compared to NLRP3KO mice infected with ΔcovRΔcylE mice and is indicated as NS; $P = 0.99$.

*et al*, 2013). In the current study, we utilized artificial lipid bilayers, RBC, and macrophages to understand how a bacterial pigment/lipid toxin disrupts these host cells. The trans-membrane conductance observed in artificial lipid bilayers suggests that the pigment/lipid toxin does not induce the formation of large, multimeric pores (commonly observed with protein toxins) but rather induces membrane permeabilization by forming variably sized membrane defects similar to those previously observed with the cyclic antimicrobial peptide, Gramicidin S (Ashrafuzzaman *et al*, 2008). The predicted length of the polyene moiety of the GBS pigment is approximately 32 Å (determined using a molecular model generated by the National Cancer Institute's Online SMILES Translator and modeled using PyMol), which is similar to the average thickness of the plasma membrane at 30–40 Å (Schrodinger, 2010). Thus, we predict that insertion of the GBS hemolytic pigment may span the host cell membrane, with the rhamnose and ornithine moieties acting as polar head groups. Insertion of the GBS pigment into RBC membranes triggers membrane disruption leading to colloidal osmotic lysis (see proposed model in Fig 8).

Unlike RBC, most host cells have the capability to turnover their plasma membrane which can confer some protection against membrane-disrupting bacterial toxins (Keyel *et al*, 2011). Although such a defense mechanism may prevent the toxin from directly killing the cell, the initial membrane disruption can activate immune pathways such as the NLRP3 inflammasome. Our data demonstrate that this is the case for the GBS hemolytic pigment. After likely intercalating into the plasma membrane, the GBS pigment triggers membrane permeability leading to ion flux. In the presence of TLR signaling (Henneke & Berner, 2006), this initiates activation of the NLRP3 inflammasome leading to caspase-1 activation, increased IL-1β and IL-18 secretion, and the programmed cell death known as pyroptosis (see proposed model in Fig 8). We demonstrate that the GBS pigment is sufficient for inflammasome activation. Previous work indicated that hemolytic GBS strains activate the NLRP3 inflammasome in mouse dendritic cells (Costa *et al*, 2012) and recently, RNA was implicated in hemolysin-mediated NLRP3 activation (Gupta *et al*, 2014). However, the extraction and isolation procedures followed for purification of the GBS pigment (Rosa-Fraile

*et al*, 2006; Whidbey *et al*, 2013) should remove any contaminating nucleic acid. In addition, we did not observe the presence of nucleic acid in purified pigment samples that were analyzed directly and subsequent to RT–PCR on ethidium bromide-stained agarose gels (Supplementary Fig S8). Furthermore, NMR data of purified pigment (Whidbey *et al*, 2013) did not reveal the presence of imino protons' characteristic of RNA (Furtig *et al*, 2003). Also, we did not detect any protein on pigment samples analyzed on Sypro Ruby-stained SDS–PAGE (Supplementary Fig S8). As the purified GBS pigment, which is devoid of nucleic acids, activates the NLRP3 inflammasome and induces pyroptosis, these results suggest that GBS pigment can induce pyroptosis even in the absence of RNA. These conclusions are also supported by our observations that inactive pigment (pigment lacking the carrier molecule starch) is not hemolytic and does not activate the NLRP3 inflammasome or induce pyroptosis (Supplementary Fig S9). These data also indicate that background absorbance values from the pigment do not confound analysis of cell death or cytokines.

Activation of the inflammasome likely leads to a more robust immune response that is programmed for pathogen clearance. Indeed, the presence of the NLRP3 inflammasome was shown to be critical for bacterial clearance in a murine model of GBS sepsis (Costa *et al*, 2012). In such a model, activation of the inflammasome was clearly beneficial to the host. However, activation of a robust inflammatory response can be detrimental during pregnancy, leading to tissue damage and perinatal complications. While increased IL-1β has been associated with chorioamnionitis, fetal injury, and preterm birth (Sadowsky *et al*, 2006; Flores-Herrera *et al*, 2012; Jaiswal *et al*, 2013), mechanistic insight into microbial factors that trigger inflammasome activation and the consequence on fetal injury and preterm birth was not known. Furthermore, a large number of studies on preterm birth have utilized either purified PAMPs that trigger TLRs such as LPS or peptidoglycan (Robertson *et al*, 2006; Breen *et al*, 2012; Jaiswal *et al*, 2013), heat-killed bacteria (Equils *et al*, 2009; Filipovich *et al*, 2009), or direct infusion of cytokines into the amniotic fluid (Sadowsky *et al*, 2006; Kallapur *et al*, 2013) to ascribe the role of cytokines and other host factors in infection-associated preterm birth. However, these strategies fail to completely recapitulate the events that occur during bacterial

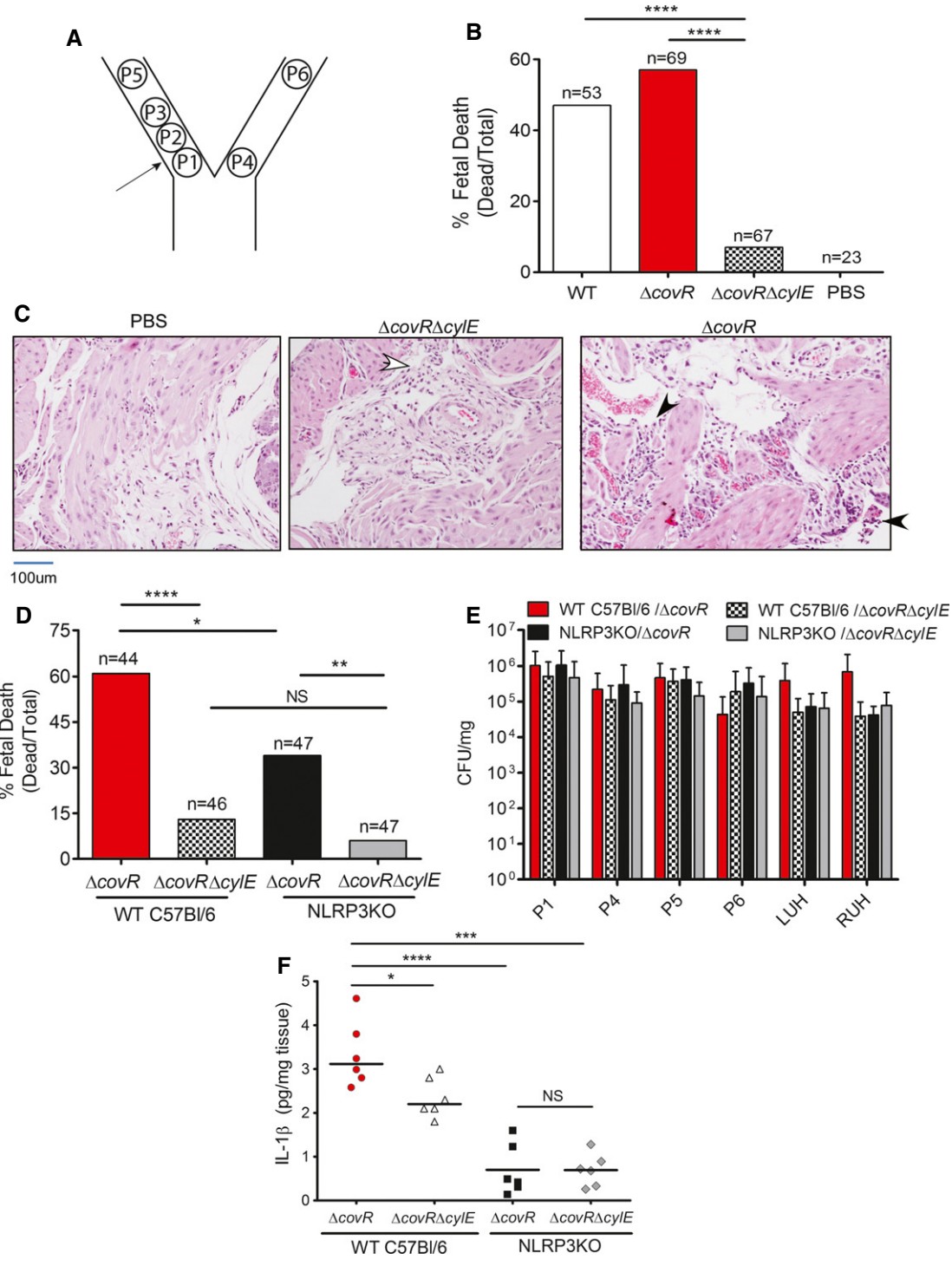

**Figure 7.**

infections *in utero*. In a murine model of *in utero* GBS infection, we observed that hemolytic and hyperhemolytic strains induce preterm birth and cause significantly more fetal death than non-hemolytic strains, similar to recent observations in a vaginal model of ascending GBS infection (Randis *et al*, 2014). Our results comparing pregnant WT and NLRP3-deficient mice suggest that pigment-mediated activation of the NLRP3 inflammasome enhances GBS infection-associated fetal injury. Thus, contrary to the benefits of inflammasome

activation during systemic GBS infection, inflammasome activation appears to be detrimental during pregnancy. The balance between these two requirements—the necessity of inflammation to protect the mother from infection and the necessity of an anti-inflammatory environment to promote fetal survival—is critical to maternal and child health.

Our studies also reveal that GBS pigment-mediated fetal injury is observed even in the absence of the NLRP3 inflammasome. These

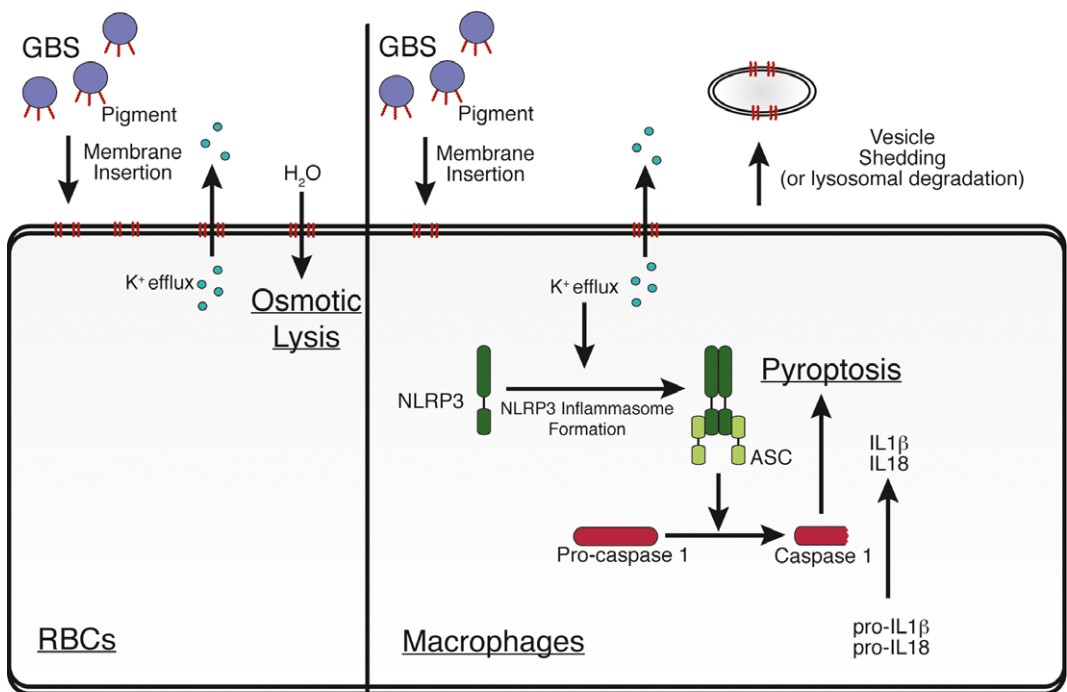

**Figure 8.  Proposed model on GBS pigment-mediated host cell lysis and preterm birth.**

The GBS pigment intercalates into host cell membranes leading to membrane permeability and ion flux. In RBC, this leads to colloidal osmotic lysis. In macrophages, the loss of intracellular potassium serves as a trigger for NLRP3 inflammasome formation. The activated inflammasome then activates caspase 1, which cleaves pro-IL-1β and pro-IL-18 to their active forms. Finally, caspase 1 activation causes cell death by pyroptosis. While NLRP3-dependent pyroptosis is the major form of cell death observed in macrophages exposed to purified pigment and hemolytic/pigmented GBS strains, hyperpigmented GBS strains induced low levels of cell death in NLRP3-deficient and ASC-deficient macrophages that were independent of inflammasome and caspase 3/7. In host cells that are either resistant or recover from pigment-induced membrane damage (e.g., NLRP3-deficient macrophages), we predict that lysosomal degradation of pigment or shedding of vesicles that contain pigment–membrane complexes and the turnover of plasma membrane prevents cell death. The GBS pigment exacerbates fetal injury and preterm birth through the combined action of colloidal osmotic lysis and pyroptosis.

data indicate that the hemolytic nature of the pigment is also likely to contribute to fetal injury. This can be expected as the pigment more efficiently lyses RBC ($EC_{50} < 0.1$ μM) when compared to IL-1β induction (0.25 μM) or pyroptosis (1–2 μM). It is also likely that inflammasome- and caspase 3/7-independent mechanisms contribute to GBS infection-associated fetal injury. Taken together, these data suggest that prevention of fetal injury and preterm birth caused by infection of hyperhemolytic GBS strains cannot solely be achieved by inhibition of inflammasome or caspase inhibitors. Rather, the generation of analogues or vaccines that prevent pigment-mediated hemolysis and pyroptosis may be necessary for prevention of GBS-associated host injury and preterm birth. Although our studies comparing GBS-induced fetal death in NLRP3KO mice and WT C57BL6J were performed using the hyperpigmented GBSΔcovR and isogenic non-pigmented GBSΔcovRΔcylE, we have previously shown that the hypervirulent phenotype of GBSΔcovR in a murine model of systemic infection (Lembo *et al*, 2010) and for penetration of human placenta (Whidbey *et al*, 2013) is reversed when the gene *cylE* important for hemolytic pigment production (Pritzlaff *et al*, 2001; Whidbey *et al*, 2013) is deleted in the GBSΔcovR mutant (Lembo *et al*, 2010; Whidbey *et al*, 2013).

Furthermore, deletion of *cylE* in GBSΔcovR did not restore the expression of CAMP factor, a CovR-activated gene (Lembo *et al*, 2010). Nevertheless, as CovR/S is a global regulator in GBS, secondary effects although unexpected are still plausible.

Given that potent nature of the pigment, it is beneficial for pathogens such as GBS, which typically exist as commensals in the lower genital tract, to repress pigment expression to minimize host injury. It is likely that under certain conditions of environmental stress (e.g., changes in pH, (Santi *et al*, 2009; Sitkiewicz *et al*, 2009)), GBS utilizes signaling systems such as CovR/S to alleviate repression of pigment genes leading to barrier disruption and ascending infection. Understanding how pathogen-encoded lipid toxins mediate host cell lysis and contribute to infection and inflammation is critical for developing preventive measures. This is relevant as opportunistic pathogens including *Bacillus cereus* contain genes homologous to those encoding the GBS hemolytic pigment/ lipid toxin (Whidbey *et al*, 2013), and other rhamnolipid toxins are produced by pathogens such as *Pseudomonas aeruginosa* and *Burkholderia pseudomallei* (Haussler *et al*, 1998; Zulianello *et al*, 2006). It is also likely that other pathogens encode hemolytic and cytolytic lipid toxins, but these are yet to be discovered. In summary, we show that the GBS

hemolytic pigment functions as a double-edged sword causing colloidal osmotic lysis or NLRP3 inflammasome activation and pyroptosis, which together exacerbate fetal injury and preterm birth. These findings emphasize the importance of cytotoxic lipids in fetal injury and have important implications for prevention of GBS pigment-mediated fetal injury and preterm births.

# Materials and Methods

## Ethics statement

Written informed patient consent for donation of human blood was obtained with approval from the Seattle Children's Research Institute Institutional Review Board (protocol # 11117) as per the Principles in the WMA Declaration of Helsinki and Dept. of Health and Human Services Belmont Report. Children under the age of 18 were not recruited for donation of human blood.

All animal experiments were approved by the Seattle Children's Research Institutional Animal Care and Use Committee (protocols #13907 and #13311) and performed in strict accordance with the recommendations in the Guide for the Care and Use of Laboratory Animals of the National Institutes of Health (8[th] Edition). All surgery was performed under anesthesia, and every effort was made to minimize suffering and animal use.

## Materials, bacterial strains, and cell lines

All chemicals were purchased from Sigma-Aldrich, unless otherwise noted. Cell culture media was purchased from Fisher Scientific. The WT GBS strains A909 and COH1 used in this study are clinical isolates obtained from infected human newborns (Lancefield *et al*, 1975; Martin *et al*, 1988). The GBS mutants Δ*cylE*, Δ*covR,* and Δ*covR*Δ*cylE* were previously derived from A909 and COH1 (Pritzlaff *et al*, 2001; Rajagopal *et al*, 2006; Lembo *et al*, 2010; Whidbey *et al*, 2013). Routine cultures of GBS were grown in tryptic soy broth (TSB, Difco Laboratories) at 37°C in 5% $CO_2$. WT THP-1s were kindly provided by Dr. Ferric C Fang. THP-1s stably transfected with shRNA to NLRP3 (C CIAS1), ASC, or scrambled controls used in this study have been previously described (Willingham *et al*, 2007; Kebaier *et al*, 2012). THP-1s were routinely cultured in RPMI with 10% fetal bovine serum as described (Willingham *et al*, 2007; Kebaier *et al*, 2012).

## Purification of the GBS pigment

GBS pigment was purified as previously described (Whidbey *et al*, 2013). Briefly, pigment was extracted from WT GBS A909 using DMSO:0.1% TFA, precipitated using $NH_4OH$ and column-purified using a Sephadex LH-20 (GE Healthcare) column as described (Rosa-Fraile *et al*, 2006; Vanberg *et al*, 2007; Whidbey *et al*, 2013). Fractions containing purified pigment were pooled, precipitated with $NH_4OH$, washed three times with HPLC-grade water, twice with DMSO, and lyophilized as described Whidbey *et al* (2013). As a control, pigment extraction protocol was performed on the GBSΔ*cylE* strain and the extract was used as a control in all experiments. The purified pigment was tested for the presence of pigment using mass spec and NMR with Δ*cylE* extract as negative controls as

described previously (Whidbey *et al*, 2013). Samples of pigment were also analyzed on protein and agarose gels to confirm the lack of DNA, RNA, and protein contaminants. For hemolytic and cytotoxic assays, lyophilized pigment or control Δ*cylE* extract was dissolved in DMSO containing 0.1% TFA and 20% starch (Difco; DTS) to a final concentration of 200 μM. The samples were incubated overnight at room temperature in the dark prior to use.

## Hemolysis assays

Hemolysis assays with purified GBS pigment were performed using human red blood cells as described previously (Whidbey *et al*, 2013). For osmotic protection assays, RBCs were pre-incubated with purified pigment for 2 min at room temperature. Cells were then pelleted by centrifugation as described (Vanberg *et al*, 2007), and the supernatant was extracted to remove unintercalated pigment. The RBCs were then resuspended in PBS or PBS containing a 30 mM solution of osmoprotectant of hydrodynamic radius 0.40 nm (PEG200), 0.56 nm (PEG400), 0.89 nm (PEG1000), 1.1 nm (PEG1500), or 1.6 nm (PEG3000) (Kuga, 1981; Vanberg *et al*, 2007) and incubated for 1 h at 37°C. Because SDS-mediated lysis occurred instantly which prevented pre-incubation with RBC, SDS was added to the resuspended RBC, which was then incubated for an hour.

## Hemoglobin and potassium efflux kinetics

To measure kinetics of hemoglobin and potassium release, human blood was centrifuged and washed three times in potassium-free PBS and resuspended to 2% RBC in potassium-free PBS. RBCs were diluted 1:1 with potassium-free PBS in a beaker and continuously stirred on a magnetic stir plate. At time 0, either 400 nM GBS pigment or an equivalent amount of control Δ*cylE* extract or 0.47 μM *Staphylococcus aureus* α-toxin (Calbiochem) was added. The concentration of free potassium in the supernatant at each time point was measured using an ion-specific electrode as per manufacturer's protocol (Cole Parmer Combination Ion Specific Electrode, EW-27504-26). Hemoglobin release was measured by centrifuging a 120-μl aliquot of the mix for 2 min at 1,200 *g* and measuring the absorbance of the supernatant at 420 nm. Percent release was calculated using the formula Percent Release = (value-minimum) / (maximum-minimum) × 100, where the minimum was the value at *t* = 0, and the maximum was the value obtained via lysis with Triton X-100 (Sanchez *et al*, 2010). A sigmoidal nonlinear regression was performed using the Prism software (GraphPad) to determine the time to 50% release.

## Conductance measurements through lipid bilayers

The single-channel conductance of the GBS pigment was analyzed on a custom-made lipid bilayer apparatus described previously (Butler *et al*, 2008). To test how the pigment alters membrane stability, we used lipid bilayers or black lipid membranes (BLMs) composed of 1,2-diphytanoyl-sn-glycero-3-phosphocholine (DPhPC; Avanti Polar Lipids) as the model. Lipid bilayers or BLMs were generated as previously described (Butler *et al*, 2008). Briefly, the reaction vessel was a Teflon block with two wells connected by a 'U-tube.' One end of this U-tube was closed except for a 20-μm aperture, which was painted with DPhPC dissolved in hexadecane. The wells

and tube were filled with buffer (0.3 M KCl, 10 mM HEPES, pH to 7.4 ± 0.05) and a pipette was used to blow and retract a 10-µl air bubble atop the lipid-painted aperture, producing the BLM as a barrier between the two chambers of the tube. A voltage (180 mV) was applied to the system, and current was measured with a patch clamp amplifier (Axon 200B). Pigment (2 µM or 75 nM) or control Δ*cylE* extract was added to one chamber, and changes in current were monitored for 10 min. As controls, BLMs were incubated with either the pore-forming toxin MspA (0.51 nM; (Butler *et al*, 2008)) or the detergent sodium dodecyl sulfate (350 µM).

### Derivation of primary macrophages from human blood

Peripheral blood mononuclear cells (PBMC) were isolated using methods previously described (Yarilina *et al*, 2011). Briefly, 60 ml of blood was collected from independent healthy human donors into heparinized tubes. The human blood was diluted 1:1 with PBS, and leukocytes were isolated using UNI-SEP maxi tubes (NOVAmed) following manufacturer's instructions. Briefly, UNI-SEP maxi tubes were spun at 1,000 $g$ for 20 min and the mononuclear cell layer was retrieved using a pipette. The isolated cells were then washed with RPMI 1640, and selection for CD14$^+$ cells was performed using the MACS magnetic separation columns (Miltenyi Biotec) as per manufacturer's instructions. Subsequently, the CD14$^+$ cells were seeded in 96-well tissue culture plates at $1 \times 10^5$ cells/well in 100 µl RPMI 160 supplemented with 10% FBS containing 100 IU/ml penicillin, 100 µg/ml streptomycin, and 20 ng/ml macrophage colony-stimulating factor (M-CSF)-1 (Life Technologies). The tissue culture media was replaced on day 5 and on day 7, and PBMC-derived macrophages were directly infected with GBS strains or primed with LPS (100 ng/ml) for 3 h before exposure to purified pigment (see below).

### Derivation of macrophages from THP-1 monocytic cells

For treatment of THP-1-derived macrophages with GBS or pigment, undifferentiated THP-1 monocytes were resuspended in media containing 100 nM phorbol 12-myristate 13-acetate (PMA) and seeded at $10^5$ cells/well into 96-well plates as described (Harrison *et al*, 2005; Holzinger *et al*, 2012). After 2 days of incubation, media were replaced with PMA-free media, were incubated overnight, and were exposed to either GBS strains or purified pigment as described below.

### Cytolysis assays

PBMC-derived macrophages and THP-1-derived macrophages from above were used to measure the cytolytic activity of the GBS strains or purified GBS pigment. For bacterial infection, PBMC- and THP-1-derived macrophages were infected with mid-log phase GBS (O.D$_{600 \text{ nm}}$ = 0.3) at an MOI (multiplicity of infection) of 1 for 4 h. Cytolysis was measured by LDH release (LDH kit, Clontech) as per manufacturer's instructions. For pigment-mediated cytolysis, cells were washed four times with PBS and treated with serum-free media containing GBS pigment (0.125–4 µM). DTS alone or Δ*cylE* extracts were used as controls. After 4 h of incubation, supernatants were removed for cytokine analysis, and the media were replaced with TC media containing serum and alamar blue (Life Technologies).

After 2 h of incubation, the fluorescence in each well was measured as per manufacturer's protocol, using a BioTek Plate Reader.

For both bacterial- and pigment-mediated cytolysis, percent of dead cells was calculated by normalizing to untreated cells (0% killing) or cells treated with 1% Triton X-100 (100% killing).

For experiments with the caspase 1 inhibitor, 50–200 µM Z-YVAD-FMK (R&D Systems), 100 µM Z-DEVD-FMK (Cayman Chemical), or equivalent amount of control DMSO were added to THP-1 cells, 1 h prior to addition of pigment or controls.

### Measurement of cytokine production

Supernatants from the cytolysis assays (see above) were used for cytokine analysis by either Luminex assay (Affymetrix) or ELISA (IL-1β: R&D Systems; IL-18: MBL) as per the manufacturer's instructions.

### Measurements of caspase-1 activation and membrane permeabilization

Activation of caspase 1 and changes in membrane permeability due to GBS pigment were measured using flow cytometry (LSRII, BD Biosciences). Briefly, THP-1 cells grown on non-TC-treated plates were washed and resuspended to $5–10 \times 10^5$ cells/ml in TC media lacking serum. For membrane permeabilization assays, uptake of the membrane impermeable dye propidium iodide (PI, Life Technologies) was measured. Briefly, THP-1 cells were treated for 30 min with 1 µM pigment or an equal volume of control Δ*cylE* extract. PI was added in the last 10 min of incubation, and uptake was measured by flow cytometry as per the manufacturer's instructions. PI+ populations were identified compared to an untreated sample. Data are representative of two independent experiments with independent preparations of pigment. For caspase 1 activation assays, cells were incubated with 1 µM pigment or an equal volume of control Δ*cylE* extract for 1 h. 660-YVAD-FMK (Immunochemistry Technologies) was then added, and incubation was continued for another hour. Caspase 1 activation was measured by flow cytometry as per the manufacturer's instructions. FLICA+ populations were identified, and percent FLICA+ cells were calculated. Data shown are representative of three independent experiments with independent preparations of pigment. Data were collected with the FACSDiva software and were analyzed using the FlowJo software system.

### Quantification of intracellular potassium concentration

Intracellular potassium concentration was measured by ion-coupled plasma–atomic emission spectroscopy (ICP-AES) as previously described (Munoz-Planillo *et al*, 2013). Briefly, THP-1 cells in 12-well plates ($10^6$ cells/well, seeded in 1 ml) were treated as described above with 1 µM pigment. At various time points (0, 30, 60, 120, 180, 240 min), supernatant was removed and wells were washed with 1 ml potassium-free PBS. Subsequently, 1 ml of 3% HNO$_3$ was added to the wells to free intracellular potassium and incubated for 30 min. One millilitre of HPLC-grade water was then added, and samples were analyzed by ICP-AES (Perkin Elmer Optima 8300). Percent intracellular potassium was calculated relative to an untreated sample.

## Murine model of intrauterine GBS infection

Wild-type CD-1-, C57BL6/J-, or NLRP3-deficient mice (Brydges *et al*, 2009; Kovarova *et al*, 2012) were used. Mice were obtained from Jackson Laboratories or Charles River Laboratory, USA, and were housed at the Seattle Children's Research Institute Vivarium in accordance with the Guide for the Care and Use of Laboratory Animals of the National Institutes of Health (8[th] Edition). Mice at 6–10 weeks of age were bred in-house for time-mated pregnancy. During mating, a female was paired individually with a male. Infected pregnant mice were housed individually. Intrauterine infections of pregnant female mice were performed as previously described (Hirsch *et al*, 1995; Elovitz *et al*, 2003) with a few modifications. Briefly, on day E14.5 of pregnancy, dams were anesthetized using isoflurane, and a midline laparotomy was performed to expose uterine horns as described previously (Hirsch *et al*, 1995; Elovitz *et al*, 2003). Bupivacaine (1–2 mg/kg) was infused along the incision site to provide local anesthesia. Dams were infected with 100 μl of $10^{6–7}$ CFU of GBS into the right uterine horn between the first (P1) and second (P2) fetal sacs most proximal to the cervix (also see Fig 7A). After inoculation, sterile saline was applied to the exposed uterus, and the uterus was returned to the abdomen. The muscle and skin layers were closed using absorbable suture as described (Hirsch *et al*, 1995; Elovitz *et al*, 2003), and dams were returned to their cage and monitored every 12 h for signs of preterm delivery. Three days post-infection or at the onset of preterm birth (vaginal bleeding and pups in cage), animals were placed under deep isoflurane anesthesia and repeated laparotomy was performed. Preterm birth was defined as delivery of at least one pup before this time point. *In utero* fetal death (IUFD) was identified by white discoloration and fetal resorption, and % fetal death was calculated using the formula: (# of dead fetuses) / (total # of fetuses) × 100. Fisher's exact test was used to compare statistical differences in fetal death by comparing number of dead to number of live fetuses obtained from six pregnant mice in each group. Bacterial load in fetus (combined CFU obtained from placenta and entire body of each fetus) was enumerated using serial dilution and plating homogenized tissues. Cytokine IL-1β levels were examined in supernatants of homogenized tissues from placenta and fetus using Luminex assays. Portions of the right uterine horn were processed for histology as described previously (Whidbey *et al*, 2013). All experiments are accurately reported as per ARRIVE guidelines.

## Statistical analysis

Student's *t*-test, Mann–Whitney test, Bonferroni's multiple comparison test following ANOVA, or Fisher's exact test were used to estimate differences as appropriate, and *P*-value < 0.05 was considered significant. The nonlinear regression analysis followed by an extra sum-of-squares F test was used to estimate and compare time to 50% release, and *P*-value < 0.05 was considered significant. These tests were performed using GraphPad Prism version 5.0 for Windows, GraphPad Software, USA, www.graphpad.com. All *in vitro* experiments were performed a minimum of three independent times in triplicate and analyzed for significance without predetermined assumptions. Based on the assumption that GBS infection will cause fetal injury, animal studies were performed with an *n* = 6 per

**The paper explained**

**Problem**

Currently, the incidence of preterm birth and prematurity in humans is at 30% in the United States and is much more severe in developing countries. Preterm birth is the largest risk factor for childhood morbidity and mortality. During human pregnancy, infection of the amniotic fluid by microorganisms residing in the lower genital tract leads to *in utero* fetal injury, preterm birth, and stillbirth. Rates of preterm births and stillbirths are on the rise in all parts of the world. Currently, there is no effective therapy to prevent preterm births or stillbirths or reduce *in utero* fetal injury. This is in part due to the significant lack of information on the nature of microbial factors and host immune responses that exacerbate human pregnancy-associated infections. Recently, we described that increased expression of the hemolytic pigment promotes GBS penetration of human placenta. However, mechanisms by which the pigment/lipid toxin causes hemolysis and host cell lysis and consequence of fetal injury and preterm birth were unknown. In the current manuscript, we address these outstanding questions.

**Results**

Here, we show that the GBS hemolytic pigment/lipid toxin induces small and transient membrane defects in artificial lipid bilayers, accumulation of these defects results in dissolution of the membrane. Pigment-induced membrane defects trigger potassium efflux resulting in osmotic lysis of red blood cells (RBCs). While these behaviors are commonly associated with pore-forming toxins, membrane permeability observed in artificial lipid bilayers demonstrates the lack of uniformly sized discrete ion channels in pigment-treated membranes. Thus, the GBS pigment shows characteristics of membrane disruption that is neither typical of pore formation nor rapid dissolution of membranes typically observed in detergents. We also show that in other eukaryotic cells (such as human macrophages), the GBS pigment induces the programmed cell death known as pyroptosis. In the absence of the NLRP3 inflammasome, pigment-mediated membrane damage and potassium efflux are observed, but cells recover from this initial damage and do not undergo pyroptosis. We further demonstrate the *in vivo* relevance of membrane permeabilization versus inflammasome activation by performing studies on *in utero* fetal death and preterm birth using WT and NLRP3 knockout mice. These data show that the GBS pigment utilizes a dual mechanism of action wherein both inflammasome activation and the hemolytic/membrane damaging nature of the pigment contribute to fetal injury. Collectively, these studies provide novel mechanistic insight into how a bacterial lipid toxin produced by GBS induces membrane permeabilization, inflammation, cell death, and fetal injury.

**Impact**

This study demonstrates how a toxic bacterial lipid can induce hemolysis and pyroptosis and how these contribute to *in utero* fetal injury and preterm birth. While previous studies have indicated that the inflammasome is important for eradication of systemic bacterial infections, here we show that activation of the inflammasome, in part, exacerbates GBS fetal injury and preterm birth. Given that GBS fetal injury is also seen in inflammasome-deficient mice, our data also suggest that mechanisms that solely prevent caspase activation (caspase I or pancaspase inhibitors) will not be sufficient to prevent GBS preterm birth as the hemolytic nature of the toxin also contributes to fetal injury. These findings have important implications in strategies for prevention of GBS infection-associated fetal injury and preterm births.

group to provide an estimated statistical power ranging from 67 to 82% depending upon standard deviation at an alpha level of 5% (~95% confidence level). Animals were randomly paired for pregnancy

and infection. No bias was introduced for infection, that is, animals were not pre-chosen for infection with a particular GBS strain. Analysis of fetal death was performed in an unbiased manner. Histology sections were scored in a blinded fashion.

**Supplementary information** for this article is available online: http://embomolmed.embopress.org

## Acknowledgements

We thank Dr. Ferric C. Fang for providing THP-1 macrophages and Dr. Jenny P-Y Ting for providing the shRNA transfected THP-1 macrophages. We thank Dr. Ida Washington for assistance with the murine model of intrauterine infection and Dr. Adrienne Roehrich for her assistance with ICP-AES. We are grateful to Drs. Brad Cookson, Ferric C. Fang, Joseph Mougous and David Sherman for helpful discussions. We are grateful to the human subjects who participated in this study. We thank members of the Rajagopal and Gundlach laboratories for technical assistance and input. We also thank Joyce Karlinsey and Jeffrey Myers for their assistance. This work was supported by funding from the National Institutes of Health, Grants R01AI100989 to LR and KAW, R56AI070749, R01AI112619, and R21AI109222 to LR, and R01AI088255 and Burroughs Wellcome Fund Career Award for Medical Scientists to JAD. CW and JV were supported by the NIH training grant (T32 AI07509, PI: Lee Ann Campbell). CW was also supported by UW GO-MAP Fellowship and JMS was supported by UW PREP NIH NIGMS 5R25 GM086304. The content is solely the responsibility of the authors and does not necessarily represent the official views of the National Institutes of Health.

## Author contributions

CW, JV, CG, EB, JMS, KD, LN, and EADE performed the experiments; JAD and JHG provided reagents; and CW, JV, CG, EB, JMS, KD, LN, EADE, JHG, MAE, DL, JAD, KAW, and LR designed the research, analyzed the results, and wrote the paper.

## Conflict of interest

The authors declare that they have no conflict of interest.

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
