## [Review Process File · EMBO Molecular Medicine]

A streptococcal lipid toxin induces membrane permeabilization and pyroptosis leading to fetal injury

Christopher Whidbey, Jay Vornhagen, Claire Gendrin, Erica Boldenow, Jenny Mae Samson, Kenji Doering, Lisa Ngo, Ejiolor A.D. Ezekwe Jr., Jens H Gundlach, Michal A Elovitz, Denny Liggitt, Joseph A Duncan, Kristina Adams Waldorf, and Lakshmi Rajagopal

Corresponding author: Lakshmi Rajagopal, University of Washington and Seattle Children's Research Institute

Review timeline:

Submission date:	19 November 2014
Editorial Decision:	14 December 2014
Revision received:	14 January 2015
Editorial Decision:	26 January 2015
Accepted:	30 January 2015

Transaction Report:

Editor: Céline Carret

1st Editorial Decision

14 December 2014

Thank you for the submission of your manuscript to EMBO Molecular Medicine. We have now heard back from the two referees whom we asked to evaluate your manuscript. Although the referees find the study to be of potential interest, they also raise a number of concerns that need to be addressed in the next version of your article.

As you will see from the comments below, both referees are rather positive about your study but still have some concerns and provide constructive suggestions to strengthen the findings. I would strongly encourage you to address all issues as recommended as we feel that this would improve the study.

Please note that it is EMBO Molecular Medicine policy to allow only a single round of revision and that, as acceptance or rejection of the manuscript will depend on another round of review, your responses should be as complete as possible.

I look forward to seeing a revised form of your manuscript as soon as possible.

***** Reviewer's comments *****

Referee #1 (Comments on Novelty/Model System):

I have only minor concerns as detailed in my comments to the authors. The only aspect of the statistics I was unsure about is in comment 1. Otherwise, this was quite an interesting piece of work.

Referee #1 (Remarks):

This manuscript by Whidbey et al. substantially builds on our knowledge of the GBS lipid toxin both in terms of mechanism of action and its role in pathogenesis. The authors convincingly demonstrate that the membrane disrupting properties of the toxin that subsequently trigger inflammasome activation. In the absence of inflammasome activation, the cells were protected from death even in the face of membrane disruption. The authors extended these findings to an *in vivo* model of preterm birth and fetal death. The manuscript is written in a clear concise manner, appropriate for a general audience. Overall, the conclusions are well supported by the data and the methodology is sound. I have only minor points for clarification that the authors may wish to consider as detailed below.

Minor Points.

1. I was a bit unsure of the statistics used for the time to release data, the extra sum-of squares F test. I am unfamiliar with this test, but it seemed that obtaining p-values of $p < 0.001$ with only 2-3 replicates (assume this would be $n=2-3$) would be challenging? Might be useful to reference another study that has used this test for this analysis or provide a brief justification for its use.
2. Page 5, lines 30-31- would be useful to show either the SDS or Triton data as an additional panel in the supplemental material.
3. The proposed model (Figure 7, page 14, line 21) was not included in the Figures. Recommend adding this file, as it will aid the general reader.
4. They mention on pg 10, line 14, that activation of NLRP3 by K efflux leads to activation of caspase 1, but they have no reference to support this.
5. Some of the figures need to be corrected. In figure 2, the order of the mutants is different in A than B. In figure 5A, there is no key to tell which lines are which. I think they need to explain the difference between Figure 1D and S3 - the main difference is the concentration of pigment, but it took me a little bit to understand why the time to disrupt the membrane was different. The order of strains in Figure 6D is odd. I think they are trying to put the two covR mutants next to each other, but it makes more sense to do the two mutants in the same order for each siRNA treatment.
6. Another good experiment to further substantiate their data in Figure 5B to demonstrate that caspase 1 is activated by the pigment would be to measure the dose-dependency of a caspase-1 inhibitor. This would be helpful since their caspase 1 inhibitor only inhibits about 50% of cell death. If the primary way the pigment is killing the cells is via caspase-1 activation, you should see less death with more caspase 1 inhibitor. However, if the pigment also causes death by pore formation-as some of their other data suggests- then you should reach a plateau after increasing caspase-1 inhibitor.
7. For figure 4C, did the authors normalize for the number of cells? They say that the NLRP3 is not present in the siRNA-treated cells and thus cell death is not induced. But, if there are more cells, and you normalize for the total number of cells, then the intracellular K might not be higher- which then might be that than the absence of NLRP3 does not prevent cell death?
8. They didn't really explain why they use two different mice in 6B and 6D. I assume it was because they had the NLRP3KO in the BL6 mice? Then why didn't they do experiment in 6B with BL6 as well? It would be helpful in panels D and F to put the mutants in the same order for the 2 different mouse strains.
9. Although they have a good control for their double mutant (the single covR mutant), the double mutant could still have some unknown secondary affects not related to cyle-especially since covR is

part of a two-component system that regulates multiple systems. This point should at least be made in the discussion.

Referee #2 (Comments on Novelty/Model System):

Effects are seen in THP-1 cells. It would be useful for the authors to verify the basic results in primary macrophages.

Referee #2 (Remarks):

Whidbey et al. report that the group B Streptococcus GBS pigment has membrane-permeabilizing activity, which can lead to efflux of K⁺ and consequent NLRP3 inflammasome activation and caspase 1-dependent pyroptosis. They show convincingly that the purified GBS pigment (as stabilized by starch) lyses red blood cells through an osmotic mechanism and that this lytic effect can be overcome by osmoprotectants (i.e., PEG 1500 and 3000), and that the purified pigment has activity on black lipid membranes. They then go on to show a second activity of the pigment either in purified form or as expressed by GBS on differentiated THP-1 cells, a tissue culture model for macrophages. The GBS pigment also causes cell death here, and a significant percentage of this death is associated with K⁺ efflux (as shown directly by measuring K⁺ levels) and dependent on NLRP3 and ASC (as shown by shRNA targeting these proteins; and protection afforded by blocking caspase 1 through the inhibitor YVAD). Lastly the authors provide evidence in a mouse model of disease that the presence of both the GBS pigment and host NLRP3 is detrimental to the host (as measured by fetal death in wild-type and NLRP3-deficient backgrounds), which is consistent with GBS pigment-mediated loss of integrity of membranes, K⁺ efflux, activation of NLRP3, and consequent pyroptosis. However, they also find that in the absence of NLRP3, the GBS pigment still exerts a substantial effect on fetal death. The mechanism for the NLRP3-independent effect is unexplained. Overall, the paper is solid and provides an important contribution, but several issues should be addressed by the authors:

1. The authors use THP-1 cells. Do primary macrophages respond in the same manner to the GBS pigment?
2. Is the unexplained NLRP3-independent effect due to osmotic lysis of cells? Did the authors try to protect macrophages with osmoprotectants to ask whether the 25-40% cell death seen in Figure 3A for *covA* in the shNLRP3 or shASC background could be eliminated or decreased?

Minor issues:

1. Figure 7 is referred to but missing.
2. The scrambled shRNA seems to have an effect compared to empty vector. This deserves some comment. And which sequence is scrambled, NLRP3 or ASC?
3. The order of samples changes between cell death and IL-1beta panels, causing some confusion.
4. The legend to Figure 4C claims that data for both GBS pigment and the *cyle* extract are shown, but the panel shows data only GBS pigment.
5. The legend to Figure 5A needs to indicate what the grey and red represent.
6. p.19, l. 23: "phorbol-12-mysteric acid" is presumably "phorbol 12-myristate 13-acetate."
7. There should be a space between numbers and units.

1st Revision - authors' response

14 January 2015

Referee #1 (Comments on Novelty/Model System):

I have only minor concerns as detailed in my comments to the authors. The only aspect of the statistics I was unsure about is in comment 1. Otherwise, this was quite an interesting piece of work.

Authors Response: We thank the reviewer for their kind words and their appreciation of our work. The issue regarding the statistics is addressed below under comment 1.

Referee #1 (Remarks):

This manuscript by Whidbey et al. substantially builds on our knowledge of the GBS lipid toxin both in terms of mechanism of action and its role in pathogenesis. The authors convincingly demonstrate that the membrane disrupting properties of the toxin that subsequently trigger inflammasome activation. In the absence of inflammasome activation, the cells were protected from death even in the face of membrane disruption. The authors extended these findings to an in vivo model of preterm birth and fetal death. The manuscript is written in a clear concise manner, appropriate for a general audience. Overall, the conclusions are well supported by the data and the methodology is sound. I have only minor points for clarification that the authors may wish to consider as detailed below.

Authors Response: We thank the reviewer for their appreciation of our work and we have addressed all the issues mentioned below. We hope these alleviate the reviewers concerns.

Minor Points.

1. I was a bit unsure of the statistics used for the time to release data, the extra sum-of squares F test. I am unfamiliar with this test, but it seemed that obtaining p-values of $p < 0.001$ with only 2-3 replicates (assume this would be $n=2-3$) would be challenging? Might be useful to reference another study that has used this test for this analysis or provide a brief justification for its use.

Authors Response: We appreciate the reviewer's comment. To analyse these data and calculate the time to 50% release, we utilized a four-parameter non-linear regression. In order to compare the EC₅₀ values (in our case, time to 50% release), we utilized the extra sum of squares F test because it is the recommended test for multiple regression analysis. The references indicated below have used the same test for EC₅₀ calculations.

<http://www.sciencedirect.com/science/article/pii/S0028390814003037>;
<http://www.jbc.org/content/287/50/41595.full>

This test is often used in pharmacology to statistically test data that has been modelled by non-linear regression.

We also used a non-linear regression analysis on each independent experiment ($n = 6$) and obtained the time to 50% release or EC₅₀ value for each of them and then utilized a paired student's t-test to compare the time to 50% release of potassium and hemoglobin and these data are significant ($p = 0.0039$). However, the t-test only utilizes the best-fit EC₅₀ values for each replicate, and does not include all of the other data from the experiment (*i.e.* overall deviations between replicates, etc.). By generating a single model using all replicates for each potassium and hemoglobin and comparing these curves by F-test, these data are taken into account.

<http://www.graphpad.com/faq/file/Prism4RegressionBook.pdf> provides a well written summary comparing the two types of analysis.

We hope the above statements address the reviewers concern.

2. Page 5, lines 30-31- would be useful to show either the SDS or Triton data as an additional panel in the supplemental material.

Authors Response: We appreciate the reviewer's comment and this has been included in Supplemental Figure 1B.

3. The proposed model (Figure 7, page 14, line 21) was not included in the Figures. Recommend adding this file, as it will aid the general reader.

Authors Response: We sincerely apologize for this omission and the model figure is now included as Figure 8.

4. They mention on pg 10, line 14, that activation of NLRP3 by K efflux leads to activation of caspase 1, but they have no reference to support this.

Authors Response: We appreciate the reviewer's comment and have included references to two papers and two review articles.

5. Some of the figures need to be corrected. In figure 2, the order of the mutants is different in A than B. In figure 5A, there is no key to tell which lines are which. I think they need to explain the difference between Figure 1D and S3 - the main difference is the concentration of pigment, but it took me a little bit to understand why the time to disrupt the membrane was different. The order of strains in Figure 6D is odd. I think they are trying to put the two covR mutants next to each other, but it makes more sense to do the two mutants in the same order for each siRNA treatment.

Authors Response: We apologize for the confusion. We have corrected these issues as mentioned below. Of note, we have included new data on primary macrophages as Figure 2 and therefore Figure numbers are shifted by one.

- a. We believe the reviewer is referring to the previous Figure 3A and B where the order of the strains was inadvertently changed leading to some confusion; this was also mentioned by reviewer 2. We have revised these panels such that the order of the strains is the same in both panels - see new Figure 4.
- b. We apologize for the omission of the key/legend in previous Figure 5 and this is now included- see new Figure 6.
- c. The concentration of pigment used in Figure 1D and S3 is now specifically indicated in the text of the manuscript - apart from their respective figure legends; we hope this clarifies the difference.
- d. We have revised the order of strains in previous Figure 6D so that the GBS strains DcovR and DcovRDcylE infected in each mouse strain namely C57BL6/J and NLRP3KO are grouped together but in the order requested by the reviewer- see new Figure 7D.

6. Another good experiment to further substantiate their data in Figure 5B to demonstrate that caspase 1 is activated by the pigment would be to measure the dose-dependency of a caspase-1 inhibitor. This would be helpful since their caspase 1 inhibitor only inhibits about 50% of cell death. If the primary way the pigment is killing the cells is via caspase-1 activation, you should see less death with more caspase 1 inhibitor. However, if the pigment also causes death by pore formation- as some of their other data suggests- then you should reach a plateau after increasing caspase-1 inhibitor.

Authors Response: We thank the reviewer for this suggestion. We have now performed this experiment as suggested using a dose response of the caspase I inhibitor ranging from 50 μ M to 200 μ M. These data are shown in Supplementary Figure 7 which indicates that cytolysis can be completely inhibited at 200 μ M Z-YVAD-FMK, and that this inhibition is dose dependent. As the reviewer mentioned, these data further support the conclusion that the GBS pigment induces pyroptosis.

7. For figure 4C, did the authors normalize for the number of cells? They say that the NLRP3 is not present in the siRNA-treated cells and thus cell death is not induced. But, if there are more cells, and you normalize for the total number of cells, then the intracellular K might not be higher- which then might be that than the absence of NLRP3 does not prevent cell death?

Authors Response: We appreciate the reviewer's comment. We would like to mention that all wells were seeded with the same initial cell density before beginning the experiment. After the indicated time, wells were washed to remove free K⁺ and nonadherent dead cells. Remaining cells were lysed with the addition of nitric acid, and intracellular K⁺ in live cells was measured by ICP-AES; these experiments were performed as described previously (Munoz-Planillo et al, 2013). We have now updated this figure to also include cytotoxicity data for both the Scrambled and shNLRP3 cells at each time point (see Figures 5C and 5D). As mentioned by the reviewer above in their synopsis, the major point we intended to indicate in this experiment is that that the shNLRP3 cells, which do not undergo significant cell death, still undergo an initial loss of intracellular potassium.

As the data show, the initial loss of K^+ due to pigment treatment is rapid and observed in both shNLRP3 and scrambled controls within 30 min during which cell death is primarily seen only with scrambled controls and not in shNLRP3 (see revised Figures 5C and 5D). The shNLRP3 cells recover back to the same level of K^+ as untreated cells at later time points (see 120min, 180min and 240 min in Figure 5D), indicating that even in the absence pyroptosis, the GBS pigment causes membrane disruption. We hope addition of the cytotoxicity data and revision of these figures clarifies the reviewers concern and that the experiment is now more interpretable.

8. *They didn't really explain why they use two different mice in 6B and 6D. I assume it was because they had the NLRP3KO in the BL6 mice? Then why didn't they do experiment in 6B with BL6 as well? It would be helpful in panels D and F to put the mutants in the same order for the 2 different mouse strains.*

Authors Response: We appreciate the reviewers comment. Our initial choice of the use of CD1 mice was done for a few reasons as outlined below.

- i. CD-1 mice have been extensively used in studies on reproductive biology and in previous intrauterine models of infection and preterm birth (Elovitz & Mrinalini, 2004; Elovitz et al, 2003; Hirsch et al, 1995; Yang et al, 2009).
- ii. CD-1 mice breed more prolifically and pregnancy is easier to estimate (McVey, 2014; Rennie et al, 2014). Also, when compared C57BL6 mice, CD-1 mice have larger litter sizes and they exhibit better maternal weight gain and peak uterine artery blood velocity (McVey, 2014; Rennie et al, 2014) and therefore these mice was suggested as a good choice by our in house veterinarian Dr. Ida Washington who helped us standardize the in utero model of GBS infection.
- iii. Once we were convinced that the in utero pregnant murine model of infection was feasible with GBS, we began studies with knockout mice. A disadvantage of CD-1 mice is that knockouts are not available. As indicated by the reviewer, the NLRP3KO mice are derived from C57BL6 mice and therefore we used the C57BL6 mice as controls for interpretation of the data obtained from the NLRP3KO mice. Because we were interested to examine the contribution of the GBS hemolytic pigment during intrauterine infection and fetal injury and based on all the *in vitro* observations with macrophages and inflammasome activation and increased preterm birth rates and fetal death in CD-1 mice, we chose to test the GBS $\Delta covR$ and isogenic $\Delta covR\Delta cylE$ strains in NLRP3KO mice and isogenic WT C57BL6. We believe that our observation that similar rates of preterm birth and *in utero fetal death* are seen in both outbred CD-1 and inbred C57BL6 mice for the GBS strains *i.e.* $\Delta covR$ and $\Delta covR\Delta cylE$ adds rigor to the study.

The order of the strains in the panels has been adjusted. We hope the above comments clarify the reviewers concerns.

9. *Although they have a good control for their double mutant (the single covR mutant), the double mutant could still have some unknown secondary affects not related to cylE-especially since covR is part of a two-component system that regulates multiple systems. This point should at least be made in the discussion.*

Authors Response: We agree with the reviewer and these statements have been included in the discussion. We have previously shown that the GBS $\Delta covR$ mutant is hypervirulent in a murine model of systemic infection (Lembo et al, 2010) and is proficient for penetration of human placenta (Whidbey et al, 2013) and that these phenotypes are reversed when *cylE* important for hemolysin and pigment production (Pritzlaff et al, 2001; Whidbey et al, 2013) is deleted in the GBS $\Delta covR$ mutant. Also, deletion of *cylE* in $\Delta covR$ did not restore the expression of CAMP factor, a CovR activated gene (Lembo et al, 2010). However, we agree with the reviewer that secondary effects are still plausible given that CovR/S is a global regulator and this is mentioned in the discussion, as suggested.

We hope the additional data and our responses above alleviate the reviewers concerns.

Referee #2 (Comments on Novelty/Model System):

Effects are seen in THP-1 cells. It would be useful for the authors to verify the basic results in primary macrophages.

Authors Response: We appreciate the reviewers concern. We have now provided data with primary macrophages as detailed in the response to comment 1 below.

Referee #2 (Remarks):

Whidbey et al. report that the group B Streptococcus GBS pigment has membrane-permeabilizing activity, which can lead to efflux of K⁺ and consequent NLRP3 inflammasome activation and caspase 1-dependent pyroptosis. They show convincingly that the purified GBS pigment (as stabilized by starch) lyses red blood cells through an osmotic mechanism and that this lytic effect can be overcome by osmoprotectants (i.e., PEG 1500 and 3000), and that the purified pigment has activity on black lipid membranes. They then go on to show a second activity of the pigment either in purified form or as expressed by GBS on differentiated THP-1 cells, a tissue culture model for macrophages. The GBS pigment also causes cell death here, and a significant percentage of this death is associated with K⁺ efflux (as shown directly by measuring K⁺ levels) and dependent on NLRP3 and ASC (as shown by shRNA targeting these proteins; and protection afforded by blocking caspase 1 through the inhibitor YVAD). Lastly, the authors provide evidence in a mouse model of disease that the presence of both the GBS pigment and host NLRP3 is detrimental to the host (as measured by fetal death in wild-type and NLRP3-deficient backgrounds), which is consistent with GBS pigment-mediated loss of integrity of membranes, K⁺ efflux, activation of NLRP3, and consequent pyroptosis. However, they also find that in the absence of NLRP3, the GBS pigment still exerts a substantial effect on fetal death. The mechanism for the NLRP3-independent effect is unexplained. Overall, the paper is solid and provides an important contribution, but several issues should be addressed by the authors:

Authors Response: We thank the reviewer for their time and for their appreciation of our work. We have addressed all the issues mentioned below and we hope these alleviate the reviewers concerns.

1. The authors use THP-1 cells. Do primary macrophages respond in the same manner to the GBS pigment?

Authors Response: We appreciate the reviewers concern. We have now provided data with primary macrophages isolated from human peripheral blood mononuclear cells (PBMCs) and this data is shown in Figure 2A-D. We observe that similar to THP-1 cells, primary macrophages derived from PBMCs show cell death and IL-1b release with pigmented GBS strains and purified pigment (see Figure 2).

2. Is the unexplained NLRP3-independent effect due to osmotic lysis of cells? Did the authors try to protect macrophages with osmoprotectants to ask whether the 25-40% cell death seen in Figure 3A for $\Delta covA$ in the shNLRP3 or shASC background could be eliminated or decreased?

Authors Response: We appreciate the reviewers comment. We have performed this experiment in the presence of osmoprotectants (PEG1500) and the data shown in Supplementary Fig. S6 (A) indicate that osmoprotectants do not provide protection against the 25% -40% cell death seen in shNLRP3 or shASC. Furthermore, we also observed that the caspase 3/7 inhibitor DEVD also did not provide protection against this cell death (Fig. S6 (B)), suggesting an apoptosis independent mechanism. As significant IL1b release was also not observed in these cells (Figure 3B), we predict that the low level of cell death observed in inflammasome deficient cells due to hyperpigmented GBS can be attributed to an inflammasome- and caspase 3/7-independent pathway. While this requires further detailed investigation, we feel this is beyond the scope of the current manuscript.

Minor issues:

1. Figure 7 is referred to but missing.

Authors Response: We sincerely apologize for this omission and the model figure is now included as Figure 8, as also mentioned by reviewer 1.

2. *The scrambled shRNA seems to have an effect compared to empty vector. This deserves some comment. And which sequence is scrambled, NLRP3 or ASC?*

Authors Response: We appreciate the reviewers comment. Although the shRNA seems to have an effect compared to empty vector in some experiments, these are not statistically significant due to the high standard deviation and this statement is mentioned. The scrambled sequence is ASC as described ((Taxman et al, 2006; Willingham et al, 2007))- and this is now indicated in the methods section.

3. *The order of samples changes between cell death and IL-1beta panels, causing some confusion.*

Authors Response: We apologize for this inadvertent error leading to confusion- the panels have been corrected.

4. *The legend to Figure 4C claims that data for both GBS pigment and the α -cylE extract are shown, but the panel shows data only GBS pigment.*

Authors Response: We appreciate the reviewers comment and we have now included the DcylE extract data and split the figure into two panels with overlapping cytotoxicity data so that the experiments is more interpretable, (see new Figures 5C and 5D).

5. *The legend to Figure 5A needs to indicate what the grey and red represent.*

Authors Response: We apologize for this omission- the legend is now provided.

6. p.19, l. 23: "phorbol-12-myseric acid" is presumably "phorbol 12-myristate 13-acetate."

Authors Response: We thank the reviewer for catching the above and this is now corrected.

7. There should be a space between numbers and units.

Authors Response: These have been corrected

We hope the additional data provided above and our responses alleviate the reviewers concerns.

In summary, our paper shows that the GBS pigment causes membrane damage to RBC and artificial lipid membranes and induces preterm birth and fetal death in an NLRP3 inflammasome dependent and independent manner. Given that studies using the purified GBS hemolytic pigment have been lacking, we believe these data are novel and will lead the fields of GBS pathogenesis, fetal injury and preterm birth in new directions. We hope that our responses to reviewer concerns and the additional data provided address the reviewers concerns and that they feel that the results provided increase an understanding of the GBS pigment and *in utero* infections and is worthy of publication in EMBO Molecular Medicine.

We thank you for your consideration.

References Cited

Elovitz MA, Mrinalini C (2004) Animal models of preterm birth. Trends Endocrinol Metab 15: 479-487

Elovitz MA, Wang Z, Chien EK, Rychlik DF, Phillippe M (2003) A new model for inflammation-induced preterm birth: the role of platelet-activating factor and Toll-like receptor-4. Am J Pathol 163: 2103-2111

Hirsch E, Saotome I, Hirsh D (1995) A model of intrauterine infection and preterm delivery in mice. *Am J Obstet Gynecol* 172: 1598-1603

Lembo A, Gurney MA, Burnside K, Banerjee A, de los Reyes M, Connelly JE, Lin WJ, Jewell KA, Vo A, Renken CW, Doran KS, Rajagopal L (2010) Regulation of CovR expression in Group B Streptococcus impacts blood-brain barrier penetration. *Mol Microbiol* 77: 431-443

McVey AW (2014) Appendix - Reproductive Parameters of Common, Commercially Available Mouse Strains. In *The Guide to Investigation of Mouse Pregnancy*, Anne Croy B, Yamada A.T, DeMayo F. J, L AS (eds), pp 791-793. Boston: Academic Press

Munoz-Planillo R, Kuffa P, Martinez-Colon G, Smith BL, Rajendiran TM, Nunez G (2013) K(+) efflux is the common trigger of NLRP3 inflammasome activation by bacterial toxins and particulate matter. *Immunity* 38: 1142-1153

Pritzlaff CA, Chang JC, Kuo SP, Tamura GS, Rubens CE, Nizet V (2001) Genetic basis for the beta-haemolytic/cytolytic activity of group B Streptococcus. *Mol Microbiol* 39: 236-247

Rennie MY, Mu J, Rahman A, Qu D, Whiteley KJ, Sled JG, Adamson SL (2014) 16 - The Uteroplacental, Fetoplacental, and Yolk Sac Circulations in the Mouse. In *The Guide to Investigation of Mouse Pregnancy*, Anne Croy B, Yamada A.T, DeMayo F. J, L AS (eds), pp 201-210. Boston: Academic Press

Taxman DJ, Livingstone LR, Zhang J, Conti BJ, Iocca HA, Williams KL, Lich JD, Ting JP, Reed W (2006) Criteria for effective design, construction, and gene knockdown by shRNA vectors. *BMC Biotechnol* 6: 7

Whidbey C, Harrell MI, Burnside K, Ngo L, Becraft AK, Iyer LM, Aravind L, Hitti J, Waldorf KM, Rajagopal L (2013) A hemolytic pigment of Group B Streptococcus allows bacterial penetration of human placenta. *J Exp Med* 210: 1265-1281

Willingham SB, Bergstralh DT, O'Connor W, Morrison AC, Taxman DJ, Duncan JA, Barnoy S, Venkatesan MM, Flavell RA, Deshmukh M, Hoffman HM, Ting JP (2007) Microbial pathogen-induced necrotic cell death mediated by the inflammasome components CIAS1/cryopyrin/NLRP3 and ASC. *Cell Host Microbe* 2: 147-159

Yang Q, Whitin JC, Ling XB, Nayak NR, Cohen HJ, Jin J, Schilling J, Yu TT, Madan A (2009) Plasma biomarkers in a mouse model of preterm labor. *Pediatr Res* 66: 11-16

2nd Editorial Decision

26 January 2015

Thank you for the submission of your revised manuscript to EMBO Molecular Medicine. We have now received the enclosed report from the referee who was asked to re-assess it. As you will see this reviewer is now globally supportive and I am pleased to inform you that we will be able to accept your manuscript pending final editorial amendments.

Please submit your revised manuscript within two weeks. I look forward to seeing a revised form of your manuscript as soon as possible.

***** Reviewer's comments *****

Referee #2 (Remarks):

The authors have suitably addressed all the issues this reviewer had previously raised.